# When Speculation Spills Secrets: Side Channels via Speculative Decoding in LLMs

## Abstract

Deployed large language models (LLMs) often rely on speculative decoding, a technique that generates and verifies multiple candidate tokens in parallel, to improve throughput and latency. In this work, we reveal a new side-channel whereby input-dependent patterns of correct and incorrect speculations can be inferred by monitoring per-iteration token counts or packet sizes. In evaluations using research prototypes and production-grade vLLM serving frameworks, we show that an adversary monitoring these patterns can fingerprint user queries (from a set of 50 prompts) with over 75% accuracy across four speculative-decoding schemes at temperature 0.3: REST (100%), LADE (91.6%), BiLD (95.2%), and EAGLE (77.6%). Even at temperature 1.0, accuracy remains far above the 2% random baseline—REST (99.6%), LADE (61.2%), BiLD (63.6%), and EAGLE (24%). We also show the capability of the attacker to leak confidential datastore contents used for prediction at rates exceeding 25 tokens/sec. To defend against these, we propose and evaluate a suite of mitigations, including packet padding and iteration-wise token aggregation.

## 1 Introduction

Large Language Models (LLMs) have transformed natural language processing (NLP), allowing machines to generate and understand human language at an unprecedented scale (Vaswani et al., 2017; Devlin et al., 2019; Brown et al., 2020; Zhang et al., 2022; Le Scao et al., 2022; Touvron et al., 2023). LLMs typically generate text using *auto-regressive* decoding (Touvron et al., 2023; OpenAI, 2024), where the generation happens serially and each token depends on all the previous ones. Unfortunately, this serial process causes a significant bottleneck in the LLM response latency (Miao et al., 2024; Fu et al., 2024) and under-utilizes the available hardware-level parallelism, limiting token generation throughput and latency.

Speculative decoding (Leviathan et al., 2023; Chen et al., 2023; Miao et al., 2024; Spector & Re, 2023) addresses this problem without impacting model accuracy. It uses smaller models or heuristics such as retrieval or self-drafting to inexpensively generate tokens speculatively, which the larger target model verifies in parallel in a single iteration. By tuning the heuristics to maintain a high rate of correct speculations, such techniques provide $2\times$ to $5\times$ speedups in inference latency and throughput (Xia et al., 2024). These techniques are being extensively adopted in inference services deployed by companies such as Cerebras (Wang, 2024) and Google (Leviathan et al., 2024).

However, the adoption of speculation techniques is not without risks. Speculative execution in CPUs (Burton, 1985; Hennessy & Patterson, 2012), which inspired speculative decoding, has led to security vulnerabilities in processors, such as Spectre (Kocher et al., 2019) and their variants, which exploit *side-channels*, i.e. timing variations due to mis-speculations that leak secret data accessed during mis-speculations. This raises the question, does speculative decoding in LLMs also introduce new risks to privacy? In this paper, we provide a study of the privacy risks of speculative decoding in LLMs, including leakage of private user inputs.

**Problem.** We observe two key issues in speculative decoding implementations: (1) the pattern of correct and incorrect speculations of output tokens is dependent on the input, and (2) input-dependent speculation patterns can be inferred from the variations in packet sizes: upon correct speculation, more tokens generated per iteration leads to a larger packet and so observing the packet size leaks the degree of correct speculation. Consequently, an adversary capable of measuring the number of tokens generated per iteration or packet sizes can gain access to input-dependent speculation-patterns based on variation in packet sizes and leak out private input attributes, or even entire inputs and outputs.

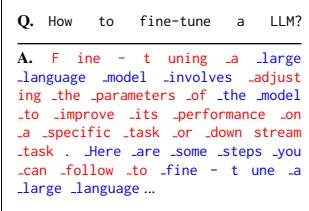

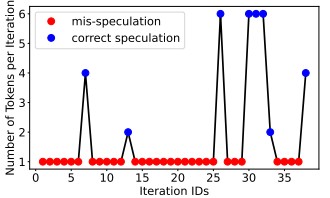

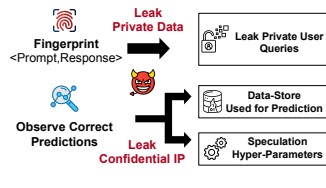

(a) Spec Decoding Example

(b) Tokens Per Iteration

(c) Potential Exploits

Figure 1: (a) In an LLM response with Speculative Decoding (e.g., LADE), tokens are either correctly speculated (blue) and verified in parallel, or mis-speculated (red) and generated via auto-regressive decoding. (b) This pattern can be inferred by measuring the number of tokens generated per iteration, where multiple tokens per iteration indicates correct speculation, and a single token per iteration indicates mis-speculation. (c) Using these patterns, a network-based adversary can fingerprint user queries and learn private user prompts and responses; a malicious user can observe correct predictions and leak out data-stores and hyper-parameters used for predictions.

Figure 1a shows a response by TinyLlama 1.1B Chat (Zhang et al., 2024a) with a speculative decoding technique, Lookahead Decoding (LADE) (Fu et al., 2024), with correct (blue) and mis-speculated (red) tokens illustrated. On correct speculation, up to $n$ tokens are verified per iteration, as shown in Figure 1b, while on mis-speculation, only one token is generated per iteration. In LLM applications streaming responses to the user, token by token, a network-based adversary, such as a malicious ISP or a compromised router (Weiss et al., 2024), can observe network packet sizes corresponding to each iteration to deduce the number of tokens per iteration and learn the input-dependent pattern of correct and incorrect speculations. We target medical chatbots (e.g., Hippocratic AI (Shah, 2025)), where users interact by asking questions about symptoms or diseases, as this is a privacy-sensitive application where any information leakage (e.g., disease or symptoms of users) breaks privacy laws.

**Exploits.** We demonstrate privacy breaches based on speculation patterns (Figure 1c). First, we show a query fingerprinting attack (e.g., leaking diseases or symptoms in medical chatbots) by a network adversary. By profiling speculation pattern fingerprints for prompt-response pairs offline, an adversary can then compare the speculation pattern for an unknown query with the fingerprints to recover private user prompts. Across speculative decoding schemes (BiLD, REST, LADE) (Fu et al., 2024; He et al., 2024; Kim et al., 2023), our prompt identification attacks (within a set of 50 queries) reaches accuracies of ∼100% for REST, 91.6% for LADE, and 95% for BiLD with a temperature of 0.3. On a remote vLLM inference server (Kwon et al., 2023), which supports speculation with EAGLE (Li et al., 2024a), we similarly observe a high accuracy of 77.6%. We further show that when the actual prompts are unavailable, the adversary can train effective fingerprints using publicly available proxy datasets; e.g., by querying a medical chatbot with the top 50 most common diseases as stand-ins for real prompts. In this case, the attack still achieves 20–40% accuracy in recovering the disease or symptoms from user prompts, far exceeding random guessing (2-6%). Lastly, we show how adversaries crafting malicious inputs can leak confidential data-store contents used for predictions with REST (He et al., 2024) at a rate of >25 tokens/second by observing correctly predicted tokens.

**Mitigations.** To mitigate these, we propose two defenses: (1) aggregating tokens over multiple iterations before transmission to obscure speculation patterns, and (2) padding packets with fixed or random bytes. Token aggregation reduces attack accuracy by up to 50% without impacting the payload size within each packet; random padding reduces it by 70% with up to 8.7× increase in the payload size. A full mitigation requires fixed size padding, which reduces attack accuracy by 98%, but increases payload size in each packet by 230×.

In summary, our contributions are as follows:

1. We observe that variation in the number of tokens generated per autoregressive iteration due to LLM speculative decoding can result in privacy breaches.

2. We demonstrate query fingerprinting attacks that can leak out exact matches of private user queries with >90% accuracy and approximate matches with 20% to 40% accuracy.

3. We also demonstrate attacks that leak confidential IP controlling the performance of the speculative decoding mechanisms, such as data from data-stores used for prediction.

4. We propose mitigations including padding packet sizes and aggregating tokens from multiple iterations to limit the potential for these exploits.

## 2 RELATED WORK

### 2.1 PRIOR SPECULATIVE DECODING TECHNIQUES

Speculative decoding improves the efficiency of auto-regressive decoding in LLMs by verifying multiple token predictions in parallel, reducing latency. These works generate tokens using a smaller model or other heuristics as predictions, which are verified by the target model.

**Speculation with Smaller Draft Models.** Prior works propose using smaller, faster draft models to generate sequences of tokens speculatively (Miao et al., 2024; Kim et al., 2023), and then verify a single sequence or a tree of such sequences in parallel using the larger target model. The draft model can be a smaller version of the model family or a pruned larger model (Yan et al., 2025). In this paper, we demonstrate our attacks on BiLD (Kim et al., 2023), as a example of this type of speculation.

**Speculation via Self-Drafting.** Recent works reuse the target model for producing predictions, eliminating the need for a separate draft model. Look-Ahead Decoding (LADE) (Fu et al., 2024) caches previous output tokens and uses them to speculate future tokens via a key-value store populated with token sequence N-grams. These are populated using Jacobi decoding, by selecting random tokens from the input and generating successive chain of tokens. When this key token reappears in the output stream, LADE uses the stored N-grams for speculation. Medusa (Cai et al., 2024) trains extra heads in the target model, each responsible for a token position in the draft sequence, for speculative tokens. EAGLE and EAGLE-2 (Li et al., 2024a;b) perform auto-regressive decoding directly on the feature layers itself to generate speculative tokens, and are integrated in the vLLM inference serving platform. Our attacks target LADE (locally) and EAGLE (on a remote vLLM server) as representative examples, but our techniques are broadly applicable to other techniques as well.

**Speculation Using Retrieval.** REST (He et al., 2024) generates predictions by retrieving data from a datastore containing relevant text or code, organized as a set of prefixes and continuations that can be potentially used as speculative tokens. If there are multiple candidates, REST organizes them in a tree, applying a tree attention mask during decoding with the target model.

**Tree-Based Verification.** SpecInfer (Miao et al., 2024), SpecTr (Sun et al., 2023) and REST (He et al., 2024) generate multiple drafts of speculation per iteration, organized in a tree structure, to increase the length of successful verification for higher speedup. These drafts are organized with a custom tree attention mask to prevent inter-tree interactions and verified in parallel. In this work, we demonstrate our attacks on REST (He et al., 2024), as a representative example of such techniques.

### 2.2 SIDE-CHANNEL ATTACKS ON LLMs

(Debenedetti et al., 2024) pioneered side-channel attacks on LLMs. Their attack exploits variation in output token counts due to the tokenizer, to leak uncommon strings in the tokenizer vocabulary. Our work instead exploits variation in token count per iteration due to speculative decoding.

(Weiss et al., 2024) identified a token-length side channel in streaming LLMs, where network attackers can infer token values by analyzing encrypted packet sizes. This attack reconstructs around 27% of the LLM outputs using the character length of each token. In our work, we similarly focus on a network-attacker observing the encrypted packet sizes, and leverage the variation in packet sizes based on speculative-decoding to identify user inputs. Moreover, we show mitigating the token-length side channel by padding tokens to the maximum token-size still results in packet size variations due to speculative decoding varying the output token counts per packet. We show that this can be used to fingerprint prompts and responses in our attacks with greater than 50% accuracy, even after the token-length side-channel is mitigated, in Appendix E.

(Carlini & Nasr, 2024) demonstrated timing attacks on LLM inference optimizations in commercial models like GPT-4 and Claude. Their attacks rely on timing variations due to these optimizations, i.e. variations in packet inter-arrival times, which is considerably susceptible to noise due to network congestion and server loads. In comparison, our attacks exploit packet size variations due to inference optimizations, which are inherently more robust and independent of server or network load variations. We show the resilience of our attacks compared to this prior work in Section 4.7.

(Song et al., 2024) and (Zheng et al., 2024) demonstrated timing side-channel attacks exploiting shared key-value (KV) and semantic caches to leak sensitive LLM inputs. Wiretapping LLMs (Soleimani et al., 2025) also reveals private attributes in LLM-based medical and financial applications.

## 3 THREAT MODEL

Like prior works (Weiss et al., 2024), we focus on *streaming* LLMs, where the LLM server sends back responses to the user token-by-token. Here, we study the following attacker models.

**Network Attacker.** We assume a malicious ISP or a compromised router that can observe network packet sizes, similar to (Weiss et al., 2024). The attacker's goal is to leak private user queries. Although LLM response packets are encrypted, the packet size is still observable. The attacker can also observe the time between each packet to identify the iteration ID. We show how an adversary can fingerprint and leak private user queries based on this in Section 4.

**Malicious User.** Here we assume the attacker is a malicious user interacting with the LLM, similar to (Debenedetti et al., 2024). The user can craft arbitrary inputs and inspect the output, and the tokens per iteration with the goal of leaking confidential data used by the LLM's speculative decoding mechanism for generating predictions. We demonstrate this attack in Section 5.

## 4 QUERY FINGERPRINTING ATTACK

### 4.1 ATTACK OVERVIEW

**Attack Scenario.** Consider a medical AI chatbot, programmed to answer a predefined set of user queries through the system prompt, such as a patient in a healthcare setting asking an LLM medical symptoms related to a disease. The queries are sent from a client device, over a network, to a server that hosts the LLM equipped with speculative decoding. This leads to the threat of a network-based attacker trying to eavesdrop on the queries and responses, as described in Section 3. Figure 2a shows the overview of our attack. Our attacker over the network seeks to learn the private query asked by the user. While the query and response packets are encrypted, the attacker can intercept the response packets, and the packet size to approximate the number of tokens per packet.

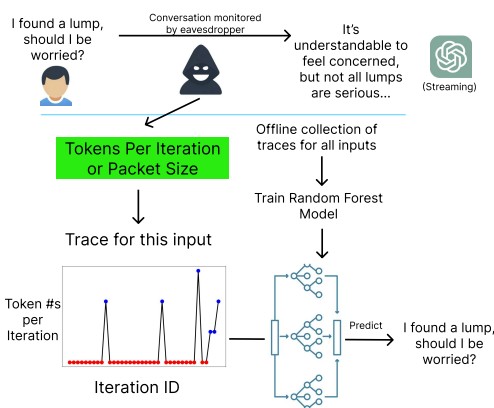

(a) Overview of the Query Fingerprinting Attack.

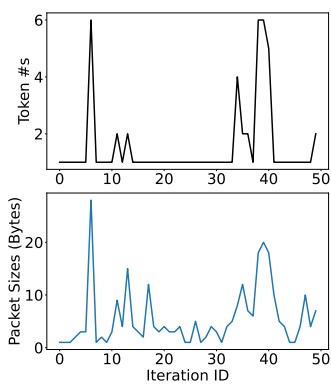

(b) Token counts and packet sizes per iteration vs. iteration-id. for a sample response with LADE.

Figure 2: (a) In the offline phase, the network-based attacker profiles variation in number of tokens per iteration influenced by speculative decoding, (based on packet sizes) and trains a classifier. In the online phase, the attacker uses the classifier to leak the input. (b) Packet sizes and tokens per iteration are correlated, allowing variations in packet sizes to be used to approximate token count variations.

**Attack Mechanism.** Since the input set is limited, the attacker fingerprints inputs based on the pattern of output token counts per iteration influenced by speculative decoding. There are two phases to the attack: offline and online. In the offline phase, the attacker profiles the number of output tokens generated per iteration vs iteration-id (we call this a *trace*) for each input in the input set, and trains a random forest classifier to predict the input based on the trace. In the online phase, the attacker collects the trace for an unknown input and uses the classifier to identify the input. Next, we provide more details about how these fingerprints are obtained, and then describe how the attack works on various speculative decoding techniques (LADE, REST, BiLD, and EAGLE).

## 4.2 Fingerprinting Using Speculation Patterns

Speculative decoding causes variation in the number of tokens generated per iteration, as correct speculation results in multiple tokens verified and sent together, while incorrect speculation generates one token per iteration. This variation can be used to fingerprint prompt/response pairs. We observe that the variations in tokens per iteration versus iteration-ID are both unique to each prompt-response combination, and are also reproducible to a great extent, as shown in Figure 8 in Appendix B.

To quantitatively validate this, we generated 2 traces each from 50 prompts taken from MedAlpaca dataset (Han et al., 2023) (Human-LLM interactions asking questions on different diseases; see Appendix A) input to LADE, with temperature of 0.3, and computed the pairwise cosine similarity between them. Traces of the same prompt (or disease) have cosine similarity of 0.9–1, whereas traces from different prompts have cosine similarity between 0.4 - 0.8. This shows that the trace of tokens per iteration with speculative decoding can uniquely fingerprint and identify a prompt and response.

As packets are encrypted, the attacker cannot see the token counts directly. However, as shown in Figure 2b, packet size strongly correlates with the number of tokens per iteration (Pearson Correlation Coefficient of 0.747 across all packets above). Thus, encrypted packet sizes serves as an effective proxy for the tokens per iteration in our fingerprinting attack. Note that exact token counts are not necessary; merely correlation of packet size with token counts suffices. Furthermore, any mitigation for prior token-size side-channels (Weiss et al., 2024) by hiding individual token sizes is insufficient, as speculation itself introduces packet size variations due to varying token counts (see Appendix E).

## 4.3 Attack Design

We attack an AI Chatbot answering medical queries from a patient. The attacker seeks to learn which query a patient asks from a finite set of queries the model can answer, to potentially leak a disease the patient has, a privacy violation by law in geographies like EU or USA. We perform three attacks:

**Experiment 1 - Exact Knowledge** The attacker has exact knowledge of the complete set of queries, allowing the attacker to profile the exact queries offline and leak them out in the online phase. We choose a set of 50 prompts from a real-world human-LLM interactions dataset (Han et al., 2023). These prompts ask a variety of questions about different diseases (the list of prompts is in Appendix A).

**Experiment 2 - Exact Knowledge (Constrained)** Next, we ask the question, do the speculation patterns depend on *structural patterns* or deeper *semantic meanings* within the response? To analyze this better, we choose 50 prompts about diseases that all starts with "What are the symptoms of" from the same dataset, ensuring all prompts have a similar structure. With this, we seek to understand whether our attack can be used to leak deeper semantic meanings associated with the prompts (e.g., diseases of the prompts) even when all the prompts have similar structures.

**Experiment 3 - Approximate Knowledge** Assuming that speculation patterns rely on semantic meanings of responses, can the attacker use this to leak private information about the prompt (e.g., user's disease) *without the exact knowledge* of the user's query? While Experiment 1 and 2 assumed training and testing prompts to be the identical, here in Experiment 3, the attacker only has approximate knowledge of user queries during profiling. The attacker's training set consists of queries that are *semantically* similar to user's test-set queries (e.g., profiled query: "For how long should you take zolpidem?" and user's query: "How long is zolpidem typically prescribed for?"). We use the prompts from Experiment 1 as our training prompts, but during testing, we rephrase the prompts using ChatGPT-4o by prompting it as "Can you rephrase the following 50 questions while keeping the content in the question the same?". We provide a more general experiment with an out-of-distribution training set in Section 4.8.

## 4.4 Experimental Setup

**Target Models.** We target four representative speculative decoding techniques: LADE (Fu et al., 2024) and EAGLE (Li et al., 2024a) using self-drafting, REST (He et al., 2024) using retrieval and tree-verification, and BiLD (Kim et al., 2023) using a smaller draft model. We run LADE with TinyLlama 1.1B Chat V1.0 (Zhang et al., 2024a), REST with Vicuna 7B (Chiang et al., 2023) as the target model and ShareGPT (Aeala, 2023) data-store of 120K conversations, and BiLD with Llama2 7B (Touvron et al., 2023) and TinyLlama 1.1B (Zhang et al., 2024a) as the target and draft model respectively. We run LADE and REST on NVIDIA RTX4090 (24GB) and BiLD on NVIDIA A100 (40GB) GPUs with Question-Answer tasks, to model a chatbot under attack.

**Datasets and Classifier.** For each experiment, we use 50 queries of varying complexity, length, and semantics (Appendix A lists the prompts). For our attack, we studied three types of classifiers - CNN, Gaussian Mixture Models (GMM) and Random Forest, and use Random Forest since it has the best accuracy. (See Figure 12 in Appendix G for the accuracy comparisons of Random Forest with CNN and GMM). For the training set in the profiling phase of the attack, we run each query 5 to 30 times (with varying temperatures) and extract the packet size trace (5 to 30 traces per query) to train our Random Forest classifier. For the test set in the online phase, we use the corresponding 50 queries and 5 new traces per query, collecting 250 data points for evaluation, and report accuracy. For our attack, we use scikit-learn's Random Forest with 150 estimators, max depth of 15, minimum 10 samples per split and 1 sample per leaf, and mean squared error as loss function.

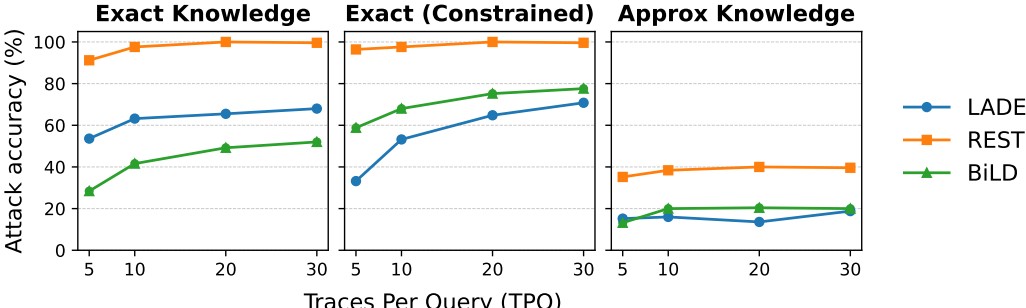

Figure 3: Accuracy of the query fingerprinting attack on LADE, REST, and BiLD with temperature of 0.8, using 5 to 30 Traces Per Query (TPQ) for training.

## 4.5 RESULTS

Figure 3 shows the results for the query fingerprinting attack on LADE, REST, and BiLD.

**Exact Knowledge Scenarios.** In both Experiment 1 and 2, the attacker leaks the input prompt with high accuracy: with accuracy of up to 70.8%, 100%, and 77.6% for LADE, REST, and BiLD respectively, compared to random guessing (2% for a set of 50 prompts). In Experiment 1, where the set of profiled and leaked prompts are identical, the attack success rates improve as the number of traces per query (TPQ) profiled increases, reaching up to 68% accuracy and an F1 score of 0.66 with 30 TPQ for LADE, reaching 99.6% accuracy with 30 TPQ and an F1 score of 1.0 for REST, and reaching 52% accuracy with 30 TPQ and an F1 score of 0.55 for BiLD. In Experiment 2, where all our prompts have a similar structure but different semantic meaning, the results are similar, with attack accuracy for LADE, REST, and BiLD reaching 90.8%, 99.6%, and 95.2% respectively. This indicates that our attack success rate is not influenced by the structure of the prompt, but relies and leaks the semantic meanings of the prompt and response.

**Approximate Knowledge Scenario.** Based on the above results, Experiment 3 uses profiled prompts that are not identical to the leaked prompts, but still semantically similar. Here, the attack success rates are lower (up to 18.8% for LADE, 20% BiLD, and 40% for REST), but still significantly better than random-guessing (2%). Thus, speculative decoding can still leak the topic of a user's prompt (e.g., user's disease), with queries that are approximately similar in semantics to the user queries. Section 4.8 details a broader out-of-distribution experiment, where the training set is taken from public sources independent of the test set.

The attacks are more successful on REST than BiLD and LADE. This is because REST has higher correct speculation rates and more stable speculation patterns (due to its reliance on retrieval from a datastore of previous conversations) compared to BiLD and LADE, enabling higher attack accuracies.

## 4.6 ABLATION STUDIES

Figure 4 presents an ablation study for the attack accuracy as the temperature of our target model varies. For LADE and BiLD, as the temperature increases from 0.3 to 1.0, the attack accuracy for both Experiment 1 and 2 decreases. This is because at lower temperatures, the generations and speculation patterns are more stable and alike. Hence, the output tokens for the same prompt and the packet sizes are more similar across runs, making the attack more successful. For REST, the attack accuracy is

stable for both Experiment 1 and 2 despite the increasing temperature. This is because the features are so pronounced that it withstands perturbations in the output caused by the increasing temperature.

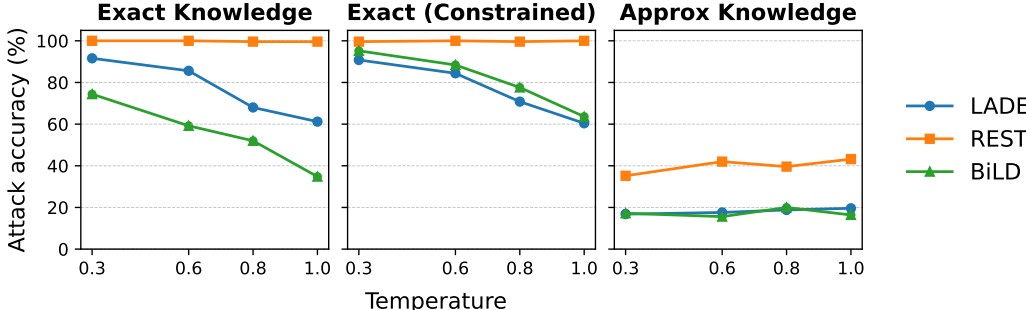

Figure 4: Attack accuracy of query fingerprinting as temperatures vary (0.3, 0.6, 0.8, 1.0), using 30 Traces Per Query (TPQ) for training.

In Experiment 3, the attacker accuracy is lower than Exp 1 or 2, as it only has approximate knowledge of the prompts (train and test set are different) However, unlike Experiment 1 and 2, for Experiment 3 the attack accuracy increases moderately as the temperature increases. This is because, at low temperatures, with limited variation in the training data, the classifier overfits on the training set, and at higher temperatures, this problem does not exist, allowing the attack be slightly more accurate.

### 4.7 ATTACK ON REMOTE vLLM SERVER

To show real-world practicality, we demonstrate our attack on a remote LLM inference server on vLLM, with Llama3 8B Instruct as the target model and EAGLE speculative decoding. Our LLM server is hosted on a cloud-based A100 GPU, while our client is located in an academic network more than 1500 miles away. Our attacker snoops the encrypted network packets in between the server and the client, using Wireshark, and captures the bytes per packet. We also compare our attack with prior work, (Carlini & Nasr, 2024), that used packet inter-arrival-times as a side-channel, that varies based on inference-time optimizations. To mimic their setup, which focused on ChatGPT, that sends tokens generated in the same iteration in separate packets, we modify vLLM to similarly send separate packets for individual tokens generated in the same iteration. For our packet size side channel, we generate our trace by combining sizes of packets received within 55 ms of each other (based on observed network latencies).

| Classifiers | Side Channel | No Server Load | | | | High Server Load | | | |
|---|---|---|---|---|---|---|---|---|---|
| | | T→ **0.3** | **0.6** | **0.8** | **1.0** | T→ **0.3** | **0.6** | **0.8** | **1.0** |
| Random Forest | Packet Size (ours) | 77.6% | 65.6% | 44.8% | 24.0% | 77.6% | 65.6% | 44.8% | 24.0% |
| | Inter-Arrival Time (Carlini & Nasr, 2024) | 14.4% | 6.0% | 5.2% | 4.8% | 6.4% | 4% | 2.4% | 2.8% |
| GMM | Packet Size (ours) | 27.2% | 7.2% | 4.8% | 4% | 27.2% | 7.2% | 4.8% | 4% |
| | Inter-Arrival Time (Carlini & Nasr, 2024) | 2.6% | 1.2% | 3.2% | 1.2% | 1.6% | 1.2% | 2.8% | 1.2% |

Table 1: Attack accuracy for the query fingerprinting (Experiment 1 - Exact Knowledge) on vLLM with EAGLE's speculative decoding at different temperatures (0.3, 0.6, 0.8, 1.0), using 30 Traces Per Query (TPQ) for training. We evaluate the attack using our packet-size side channel, and with inter-arrival times based side channel from prior work (Carlini & Nasr, 2024).

Table 1 shows that, under no additional server load, our attack achieves a maximum of 77.6% accuracy, significantly outperforming prior work which achieves a maximum accuracy of 14.4% (both using Random Forest classifier). This is because inter-arrival times provide a noisy signal due to varying network contentions. Moreover, under spikes in server load, the inter-arrival times (time per output token) can significantly spike as observed in recent work (Zhang et al., 2024b), causing test-time observations to vary dramatically compared to training time. Consequently, assuming a standard log-normal distribution of time-per-iteration to mimic high server load (Benson et al., 2009), the accuracy of prior timing channel attacks drops to 6.4%. In comparison, our attack accuracy is unaffected, since we rely on the packet size side-channel, which remains robust to variations in server loads or network congestion. Even when using GMM instead of Random Forest like prior

work (Carlini & Nasr, 2024), the trends are similar, with our packet-size side-channel outperforming prior inter-arrival time side-channel (Carlini & Nasr, 2024) under both low and high server loads.

## 4.8 OUT-OF-DISTRIBUTION TRAINING

To study the effect of distribution mismatch between profiling and attack phases, we conducted an additional experiment where the attacker trains on an out-of-distribution (OOD) set of queries. Shown in Appendix A.5, we used GPT-4o to generate 50 common diseases that users typically ask AI chatbots about, prepending the phrase "what are the symptoms of" to each. This OOD set was used for training, while evaluation was performed on the same 50 disease-related queries from Experiment 1 (with model temperature of 0.3 to 0.8), ensuring no prior knowledge of the test dataset during training.

Despite the lack of prior knowledge of the true query set, the attack substantially outperformed our previous naive random guessing attacks (∼2%). LADE achieved 23% - 25% accuracy, REST 35% - 36%, and BiLD 23% - 25%, where success was defined as predicting a disease with similar symptoms (as per GPT-5 mini) compared to the ground truth, as temperature of the target model varies between 0.3 and 0.8 (see Table 3 in Appendix F for detailed results). For comparison, we also perform a more "sophisticated" random guessing attack using OOD data: for each query in our test set, we randomly pick a disease from the OOD set as our guess, and check whether the guess has symptoms overlapping with the ground truth. This random guessing with OOD data has 6% accuracy (higher than 2% with our previous naive random guessing); this is still much lower than the 23% to 35% accuracy we achieve by training our classifier using the speculation traces observed with OOD data.

These results show that although accuracy drops relative to the in-distribution setting, speculative decoding still leaks strong and learnable signals even under OOD conditions, without being much affected by temperature, underscoring the robustness of our side-channel attack.

## 5 LEAKING PRIVATE DATA USED FOR SPECULATION

Speculative decoding techniques may use carefully tuned data to ensure high correct speculation rates, which may contain private user data or other intellectual property. In this section, we show how a malicious user can leak out private data used for predictions. We show this on REST (He et al., 2024), which relies on retrieval from a datastore to generate speculative tokens. This datastore can be populated with proprietary information to tailor model output to specific domains, or data collected from user interactions, which may be leaked by our attack. A similar attack can also leak speculation hyperparameters, which can be a company's intellectual property, as shown in Appendix C.

**Attack Design.** REST retrieves the longest matching sequence from its datastore based on the tokens generated so far and uses the subsequent continuations as speculative tokens. Any tokens that are correctly speculated, *i.e.*, returned as a group of tokens per iteration, have to exist in the datastore. We propose an attack where a malicious user crafts inputs and observes which tokens are correctly speculated and systematically leaks sequences of tokens from the datastore. We study the following attacker strategies for input generation to achieve high token leakage rates:

**1. Random Generation:** The user prompts the LLM to generate random paragraphs of text.

**2. Leveraging Common Words.** The LLM is prompted to generate paragraphs containing the most frequently occurring words, drawn from a dataset of the 10,000 most common English words (Brants & Franz, 2021). This exploits the high likelihood that phrases in the datastore contain these words.

**3. Reusing Leaked Sequences.** The LLM is iteratively prompted to generate text extending phrases already leaked out, as shown in Figure 5a. This approach is based on the insight that the sequence that is already leaked out may be a part of a longer phrase in the datastore.

**Results.** We test our attacks on REST using the ShareGPT datastore of 120K conversations for retrieval on NVIDIA RTX 4090. Figure 5b shows the unique sequences leaked by each attack strategy, averaged over three runs. Random generation initially leaks 20K sequences in 20 minutes, but slows down, leaking only 70K sequences in 3 hours. Leveraging common words improves this to 31K unique sequences leaked in 20 minutes and 190K after 3 hours. Lastly, prompting with leaked sequences is the most effective, leaking 35K unique sequences in 20 minutes and 200K in 3 hours, demonstrating how feedback amplifies leakage. These results show that a malicious user can reliably extract REST's datastore via speculation patterns.

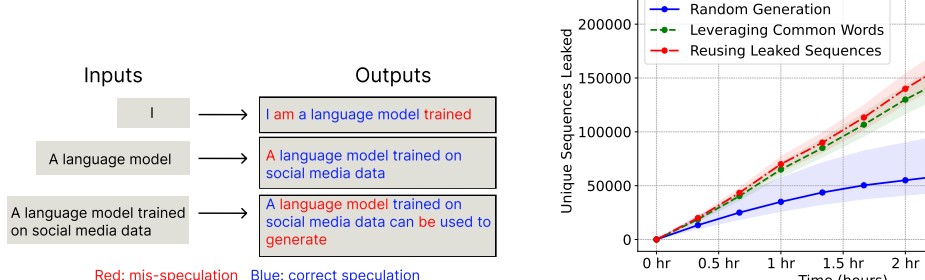

(a) Exploiting leaked sequences for longer extraction.      (b) Unique Sequences Leaked vs Time (avg of 3 runs).

Figure 5: Analysis of datastore leakage from REST.

## 6 MITIGATION STRATEGIES

### 6.1 MITIGATING QUERY FINGERPRINTING ATTACKS

To mitigate the query fingerprinting attacks, we propose (1) padding token packets and (2) aggregating tokens over multiple iterations. We describe both below:

**1. Padding Token Packets**: Padding the network packet size with additional bytes can mask the actual number of tokens per iteration and conceal the differences between correct and mis-speculations to prevent the attack. The padding can be added in two ways:

**A. Constant Length Padding.** The payload (token bytes) within each packet is padded to a constant size, selected based on the maximum number of tokens per iteration, multiplied by the maximum bytes per token. We pick this size as 1024 bytes without loss of generality, given that the maximum number of tokens per iteration can be as high as 128 in recent works (Zhao et al., 2024). This completely mitigates the side channel, reducing the attack accuracy to random guessing (2%). However, this comes at the cost of increased communication overheads, with an increase in payload size within the packet by 230× (calculated using only the token bytes and ignoring metadata, since metadata size varies across inference frameworks). This bounds the worst-case increase in packet size.

**B. Random Padding.** Packets can also be padded with a random number of bytes ($\epsilon$) selected from a uniform distribution of $\epsilon \sim Unif(0, D)$, where D is an upper bound of padding and $\epsilon$ is an integer.

Figure 6 shows the attack accuracy, as D goes from 6 to 48. As D increases to 48, the attack accuracy for LADE and BiLD drops close to random guess (2%) for all experiments. The attack accuracy for REST also shows a considerable drop from 100% without padding to 27% and 34% with padding of D=48 for Experiments 1 and 2 respectively. Similarly, for Experiment 3, attack accuracy drops from 40% to 8% with D=48. These come at the cost of a 5× to 8× increase in payload size within packets for D=48 (See Table 2 for details). Thus, variable length padding provides better reduction in attack accuracy with limited increase in payload size, compared to constant length padding.

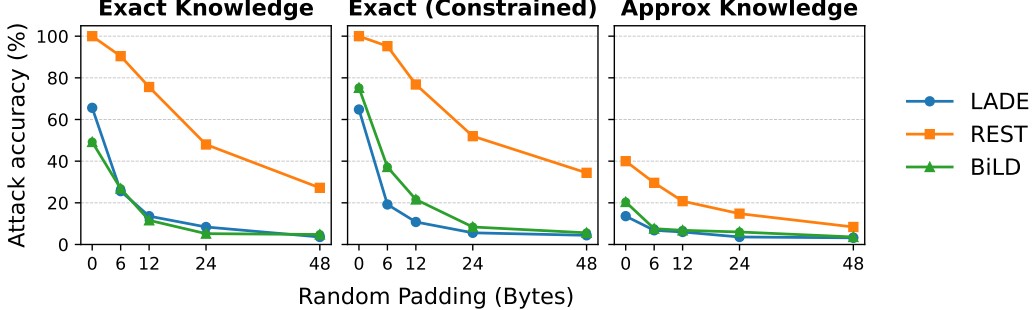

Figure 6: Attack accuracy under variable-length padding (random padding) added to payloads within packets as a mitigation. Variable-length padding follows a uniform distribution, $\epsilon \sim Unif(0, D)$, with $D$ from 6 to 48. (Attack: 20 traces per prompt; model temperature: 0.8).

**2. Reduce Granularity of Attacker Observation**: Alternatively, the LLM server can aggregate output tokens over multiple iterations and return them together to the client. This limits observability of fine-grain speculation patterns from packet sizes, reducing the success of fingerprinting attacks. However, token aggregation linearly increases the inter-arrival time for packets, which linearly increases the latency observed by a user between text renderings that may hinder user experience.

Figure 7 shows the accuracy for all our fingerprinting attacks after aggregating output tokens over 3 to 20 iterations. The accuracy of the exact knowledge attacks (Experiment 1 and 2) for LADE, REST, and BiLD drop to a minimum of 9.2%, 52%, and 11.2% respectively after aggregating tokens over 20 iterations, down from 65%, 100%, and 49.2% respectively without aggregation; accuracy of the approximate knowledge attacks (Experiment 3) decreases to a minimum of 5.2%, 17.6%, and 4% respectively, down from 14%, 40%, and 20% respectively. Thus, token aggregation is an effective mitigation. A downside of aggregating tokens over larger iterations is the increased latency between outputs, which can hinder system responsiveness. At the extreme, aggregating all tokens in a single packet resembles a non-streaming LLM.

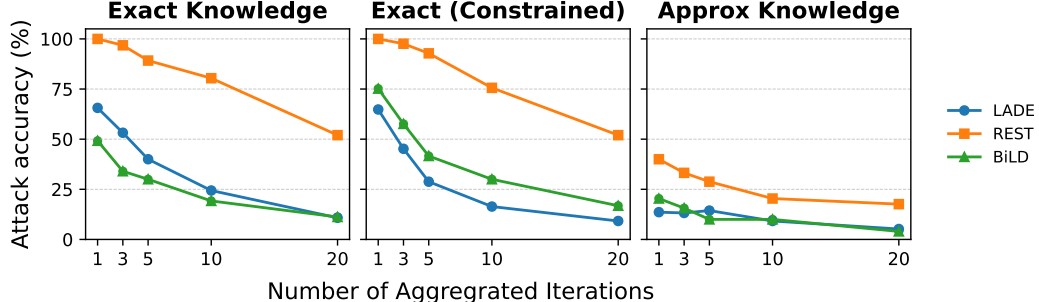

Figure 7: Token Aggregation Mitigation. Attack accuracy (30 traces/prompt, temperature of 0.8) decreases as aggregation granularity (AGG) increases, i.e., iterations tokens are aggregated over, varies from 1 (no aggregation) to 20.

**3. Packet Splitting.** Another way of limiting attacker observability of token counts, is to split the multiple tokens generated in an iteration into individual packets that are spread out in time throughout the decode iteration time period. To ensure that the attacker cannot observe even an approximate count of tokens-per-iteration by tracking the number of packets sent during one decode iteration, the defense needs to further also limit the number of packets per iteration and carry excess tokens to later iterations. This can introduce an overall latency impact, varying based on the framework, model and speculation technique. Future works can develop novel speculation techniques with such traffic shaping built in, to get both security and performance.

### 6.2 MITIGATING DATASTORE LEAKS

**Use Public Data for Speculation.** In addition to padding or token aggregation, to prevent confidential data leakage, a simple approach is to ensure any data-store used for speculation (like in REST) only contains public data, and any private or personal identifiable information (e.g., usernames or address) is anonymized. This eliminates the risk of sensitive data leakage through speculation.

### 7 CONCLUSION

This paper reveals significant privacy risks associated with speculative decoding in Large Language Models. Across multiple speculative decoding techniques, we demonstrate that attacks can exploit speculation patterns to infer user inputs with > 90% accuracy via fingerprinting attacks on local models and on models remotely hosted on vLLM inference servers, and similarly leak data from datastores used for predictions. These leaks highlight the careful consideration needed when deploying speculative decoding techniques to ensure that performance does not come at the cost of privacy.

## 8 ETHICS STATEMENT

This research investigates side-channel vulnerabilities arising from speculative decoding in large language models (LLMs). All our attacks were conducted on academic prototypes and locally hosted inference servers (e.g., vLLM). No live production systems were targeted, and no real user data was accessed or exposed. While our work demonstrates the existence of new leakage vectors, we also attempt to prevent harmful misuse. Specifically, we present mitigation strategies alongside our findings to support the development of more secure future LLM deployments. Our intent is to improve the resilience of systems rather than enable adversarial exploitation.

## 9 REPRODUCIBILITY STATEMENT

We have uploaded the source code for all our experiments in the supplementary material. Upon acceptance of the paper, we will also make the source code and instructions to run it publicly accessible, for ease of reproducibility.

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

## A PROMPTS FOR FINGERPRINTING ATTACKS

### A.1 TRAINING AND TESTING PROMPTS FOR EXPERIMENT 1

What to expect if I have Allergy (Outlook/Prognosis)?

What causes Cardiogenic shock?

What to expect if I have Congenital heart block (Outlook/Prognosis)?

What are the symptoms of Congestive heart failure?

What are the symptoms of Coronary heart disease?

What to expect if I have Dengue fever (Outlook/Prognosis)?

What causes Stroke?

Who is at highest risk for Heart attack?

What to expect if I have Heart murmur (Outlook/Prognosis)?

When to seek urgent medical care when I have Metabolic syndrome?

What are the symptoms of Palpitation?

What causes Radiation injury?

What are the symptoms of shock?

Who is at highest risk for Back pain?

What are the symptoms of Athlete's foot?

What causes Boil?

When to seek urgent medical care when I have Burns?

What causes Corns & calluses?

Who is at highest risk for Fleas?

When to seek urgent medical care when I have Hand-foot-and-mouth disease?

When to seek urgent medical care when I have Hives?

When to seek urgent medical care when I have Liver spots?

What to expect if I have Measles (Outlook/Prognosis)?

When to seek urgent medical care when I have Mumps?

What are the symptoms of a Rash?

When to seek urgent medical care when I have Ringworm?

What are the symptoms of Scarlet fever?

What causes Shingles?

What are the symptoms of Sunburn?

When to seek urgent medical care when I have Wart?

What are the symptoms of Common cold?

Who is at highest risk for AIDS ?

What are the causes of Alcoholic liver disease?

What to expect if I have Anal cancer (Outlook/Prognosis)?

When to seek urgent medical care when I have Anal fissure?

When to seek urgent medical care when I have Anemia?

When to seek urgent medical care when I have Antisocial personality disorder?

How many children have Autism?

What are the symptoms of Avian influenza?

When to seek urgent medical care when I have Avoidant personality disorder?

What causes cataract?

Who is at highest risk for Chickenpox?

What causes Chronic fatigue syndrome?

When to seek urgent medical care when I have Depression?

What to expect if I have Color blindness?

How many children have Dependent personality disorder?

When to seek urgent medical care when I have Adolescent depression?

Who is at highest risk for Diabetes?

What to expect if I have Ebola?

What causes fatty liver?

## A.2 TRAINING AND TESTING PROMPTS FOR EXPERIMENT 2

What are the symptoms of Allergy?

What are the symptoms of Cardiogenic shock?

What are the symptoms of heart block?

What are the symptoms of Congestive heart failure?

What are the symptoms of Coronary heart disease?

What are the symptoms of Dengue fever?

What are the symptoms of stroke?

What are the symptoms of a Heart attack?

What are the symptoms of Heart murmur?

What are the symptoms of Metabolic syndrome?

What are the symptoms of Palpitation?

What are the symptoms of Radiation injury?

What are the symptoms of shock?

What are the symptoms of Back pain?

What are the symptoms of Athlete's foot?

What are the symptoms of Boil?

What are the symptoms of Burns?

What are the symptoms of Corns & calluses?

What are the symptoms of Fleas?

What are the symptoms of Hand-foot-and-mouth disease?

What are the symptoms of Hives?

What are the symptoms of Liver spots?

What are the symptoms of Measles?

What are the symptoms of Mumps?

What are the symptoms of a Rash?

What are the symptoms of Ringworm?

What are the symptoms of Scarlet fever?

What are the symptoms of Shingles?

What are the symptoms of Sunburn?

What are the symptoms of Warts?

What are the symptoms of Common cold?

What are the symptoms of AIDS?

What are the symptoms of Alcoholic liver disease?

What are the symptoms of Anal cancer?

What are the symptoms of an Anal fissure?

What are the symptoms of Anemia?

What are the symptoms of Antisocial personality disorder?

What are the symptoms of Autism?

What are the symptoms of Avian influenza?

What are the symptoms of Avoidant personality disorder?

What are the symptoms of a cataract?

What are the symptoms of Chickenpox?

What are the symptoms of Chronic fatigue syndrome?

What are the symptoms of Depression?

What are the symptoms of Color blindness?

What are the symptoms of Dependent personality disorder?

What are the symptoms of Adolescent depression?

What are the symptoms of Diabetes?

What are the symptoms of Ebola?

What are the symptoms of fatty liver?

### A.3    TRAINING PROMPTS FOR EXPERIMENT 3

Same as Appendix A.1

### A.4    TESTING PROMPTS FOR EXPERIMENT 3

What is the outlook for someone with an allergy?

What leads to cardiogenic shock?

How might congenital heart block affect me in the long term?

How does congestive heart failure present itself?

How can coronary heart disease symptoms be identified?

What is the prognosis for someone with dengue fever?

What are the main causes of a stroke?

Who faces the greatest risk for a heart attack?

What is the long-term prognosis for someone with a heart murmur?

When is it critical to get help for metabolic syndrome?

How can I tell if I'm experiencing palpitations?

What factors contribute to radiation injury?

What are the warning signs of shock?

Who is most susceptible to back pain?

What should I look out for if I suspect athlete's foot?

What are the main causes of a boil?

When are burns severe enough to seek emergency medical care?

Why do corns and calluses form?

Who is most at risk of getting fleas?

When should I go to the doctor for hand-foot-and-mouth disease?

When is hives a sign to seek urgent medical attention?

When should liver spots be evaluated by a doctor?

What can I expect if I have measles?

When should I seek medical help for mumps?

How do I know if I have a rash?

When should ringworm be treated urgently?

What are the common symptoms of scarlet fever?

What is the underlying cause of shingles?

How can I recognize sunburn symptoms?

When should a wart be looked at by a doctor?

What are the signs of the common cold?

Who is most at risk of contracting HIV/AIDS?

What factors contribute to alcoholic liver disease?

What is the outlook for someone diagnosed with anal cancer?

When is an anal fissure an emergency?

When should anemia symptoms prompt immediate medical care?

When is antisocial personality disorder considered an urgent issue?

How common is autism in children?

What symptoms are typical of avian influenza?

When should avoidant personality disorder be addressed by a professional?

What are the main causes of cataracts?

Who is most vulnerable to contracting chickenpox?

What are the underlying causes of chronic fatigue syndrome?

When should I seek emergency medical help if I have depression?

What are the typical symptoms or outcomes if I have color blindness?

How common is dependent personality disorder in children?

When is urgent medical attention needed for adolescent depression?

Who faces the highest risk of developing diabetes?

What should I expect if I have Ebola?

What leads to the development of fatty liver?

## A.5 OOD TRAINING PROMPTS FOR SECTION 4.8

What are the symptoms of Common cold?

What are the symptoms of Influenza (flu)?

What are the symptoms of COVID-19?

What are the symptoms of Strep throat?

What are the symptoms of Pneumonia?

What are the symptoms of Bronchitis?

What are the symptoms of Tuberculosis (TB)?

What are the symptoms of Urinary tract infection (UTI)?

What are the symptoms of Sexually transmitted infections (chlamydia, gonorrhea)?

What are the symptoms of Lyme disease?

What are the symptoms of Type 2 diabetes?

What are the symptoms of Hypertension (high blood pressure)?

What are the symptoms of High cholesterol?

What are the symptoms of Obesity?

What are the symptoms of Chronic kidney disease?

What are the symptoms of Osteoarthritis?

What are the symptoms of Rheumatoid arthritis?

What are the symptoms of Asthma?

What are the symptoms of Chronic obstructive pulmonary disease (COPD)?

What are the symptoms of Hypothyroidism?

What are the symptoms of Coronary artery disease?

What are the symptoms of Heart failure?

What are the symptoms of Atrial fibrillation?

What are the symptoms of Stroke / transient ischemic attack (TIA)?

What are the symptoms of Peripheral artery disease?

What are the symptoms of Migraine?

What are the symptoms of Epilepsy?

What are the symptoms of Parkinson's disease?

What are the symptoms of Multiple sclerosis?

What are the symptoms of Alzheimer's disease / dementia?

What are the symptoms of Gastroesophageal reflux disease (GERD)?

What are the symptoms of Irritable bowel syndrome (IBS)?

What are the symptoms of Peptic ulcer disease?

What are the symptoms of Gallstones?

What are the symptoms of Hepatitis (A, B, C)?

What are the symptoms of Breast cancer?

What are the symptoms of Prostate cancer?

What are the symptoms of Lung cancer?

What are the symptoms of Colorectal cancer?

What are the symptoms of Skin cancer (melanoma, basal cell carcinoma)?

What are the symptoms of Depression?

What are the symptoms of Generalized anxiety disorder?

What are the symptoms of Panic disorder?

What are the symptoms of Bipolar disorder?

What are the symptoms of Post-traumatic stress disorder (PTSD)?

What are the symptoms of Eczema (atopic dermatitis)?

What are the symptoms of Psoriasis?

What are the symptoms of Acne?

What are the symptoms of Rosacea?

What are the symptoms of Fungal skin infections (ringworm, athlete's foot)?

## B  UNIQUENESS AND REPRODUCIBILITY OF FINGERPRINTS

Figure 8 shows the trace of tokens per iteration for two different prompts with LADE (temperature of 0.3) across different runs. By inspecting these t races, we observe two key properties that make these traces suitable for usage to identify and leak the prompt and response:

1. **Traces are unique to a prompt and response** - different prompts result in different traces of tokens per iteration.

2. **Traces are reproducible** - repeating the same prompt produces similar patterns in the trace of token counts.

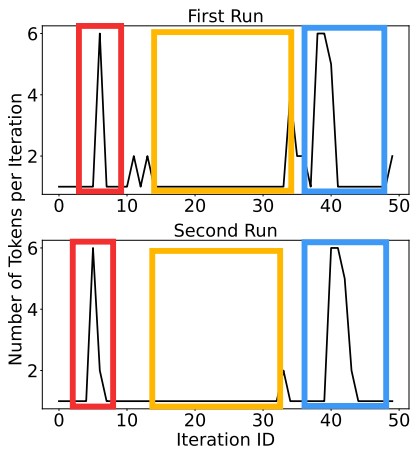
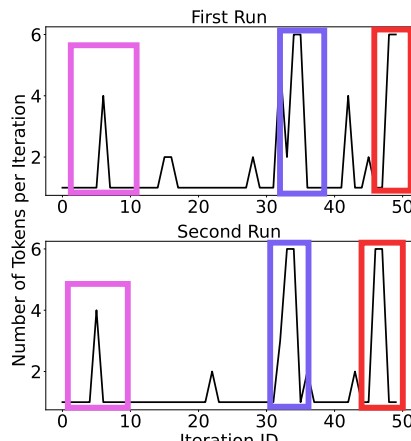

(a) What is narcissistic personality disorder?  (b) What are the symptoms of cancer?

Figure 8: Trace of Tokens per iteration vs Iteration-ID, for two runs of different prompts (a) and (b) with LADE. The trace has **unique** and **reproducible** patterns, enabling its usage as a fingerprint for leaking prompts.

## C    LEAKAGE OF SPECULATION HYPER-PARAMETERS

Speculative decoding mechanisms rely on hyper-parameters to control speculation, directly impacting correct speculation rates and performance. By analyzing traces of these patterns, we can reverse-engineer the hyper-parameters used. We demonstrate this attack on LADE Fu et al. (2024).

### C.1    BACKGROUND ON LADE'S IMPLEMENTATION

LADE generates speculative tokens by caching and reusing prior n-grams generated by the model. LADE has two main parameters: N for n-gram size, and G for guess set size. Its cache is structured as a key-value store, where the key is a token, and the value is a list of G candidate (N-1)-grams following the key token, that are used for speculation. These are replaced in the Least-Recently-Used (LRU) order. During execution, LADE use the last token of the previous iteration or the input to query this cache, and the associated G candidate (N-1)-grams are used as speculation candidates, and the best match with target model generation is accepted.

### C.2    LEAKING HYPER-PARAMETER N IN LADE

As each iteration outputs at most N-1 tokens under correct speculation, we expect the maximum number of tokens per iteration to be N-1. Therefore, we prompt LADE with a simple prompt, "Repeat Letter 'A' 60 times" that should sustain the maximum correctly speculated tokens per iteration. We observe the maximum number of tokens per iteration with this prompt to deduce N-1, and learn N.

Figure 9a shows the number of correct speculated tokens for LADE with three different values of $N$ for this prompt. We see that for each value of N (4, 5, and 6), this prompt has a maximum of 3, 4, and 5 tokens per iteration (N-1), letting us learn the value of N as 4, 5, or 6 in each of these cases. This technique can be extended to learn any value of N used in LADE.

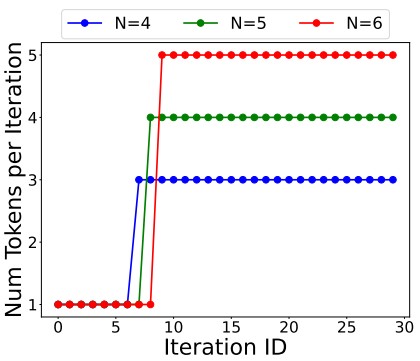

(a) Tokens per iteration with LADE for different $N$, with a prompt that achieves maximum correct speculation ("Repeat Letter 'A' 60 times"). The maximum tokens per iteration corresponds to N-1.

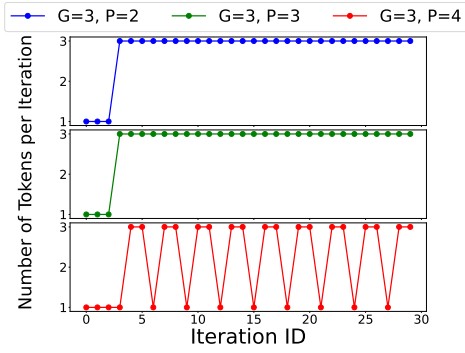

(b) Tokens per iteration with LADE with Guess Set (G = 3), using a prompt that asks to repeat P phrases sharing a common prefix "I run". Periodic mis-speculations occur after "run", only when $P > G$, leaking the value of $G$.

Figure 9: Leaking Speculation Hyper-parameters of LADE.

### C.3    LEAKING HYPER-PARAMETER G IN LADE

To leak G, the number of candidate predictions considered for speculation, we force LADE to incur targeted mis-speculations. Here, we prompt the model to repeat a sequence of phrases with the same prefix ("I run"), i.e., "I run to the grocery shop. I run as fast as cars. I run with my best friend. I run from the lecture hall.". If the number of such phrases is less than or equal to G, LADE caches all the N-grams following the key "run", resulting in it being able to correctly speculate the token that follows "run" in each of these phrases. However, if the number of phrases are greater than G, then with LRU replacement, newer n-grams evict older n-grams for the key "run" from the set of candidates which has a limited capacity of G. This causes the token following "run" to always be mis-speculated.

Figure 9b shows the traces of token generation time for LADE ($G = 3$), as the number of phrases ($P$) of the type "I run ..." increases from 2 to 4. When $P$ is 2 or 3 ($\leq G$), we notice all the tokens

after the initial ones are correctly predicted in the generation (high number of tokens per iteration). But when $P$ is 4 ($> G$), every 7th token, i.e. the token following "run" is mis-speculated (one token per iteration). Thus, we leak the value of G to be 3. This attack can be generalized to any value of parameter G in LADE.

## D  RESULTS FOR RANDOM PADDING MITIGATION

Table 2 shows the impact of constant-size padding added to payloads and padding random bytes to the payload within the packets. We show both the impact to the accuracy of the attack and the increase in the payload size within packets (token bytes). Across all attacker knowledge settings, no padding allows high attack accuracy (up to 100%). Fixed padding (1024 bytes) consistently reduces accuracy to within 2–4% but can have a high overhead in the payload size (increasing token bytes by 230×). Random padding (D=48) suppresses accuracy to a minimum of 4.4-34.4% across models at reasonable overhead (5×–8×).

| | No Padding | Fixed Padding (1024 bytes) | Random Padding (bytes) | | | | Payload Size Overhead | | | |
|---|---|---|---|---|---|---|---|---|---|---|
| | | | D → 6 | 12 | 24 | 48 | D → 6 | 12 | 24 | 48 |
| **Experiment 1 - Exact Knowledge** | | | | | | | | | | |
| LADE | 65.6% | 3.2% | 25.6% | 13.6% | 8.4% | 3.6% | 1.5× | 2× | 3.1× | 5.1× |
| REST | 100% | 2% | 90.4% | 75.6% | 48% | 27.2% | 1.9× | 2.8× | 4.6× | 8.2× |
| BiLD | 49.2% | 2% | 26.8% | 11.6% | 5.2% | 4.8% | 1.7× | 2.3× | 3.6× | 6.3× |
| **Experiment 2 - Exact Knowledge (Constrained)** | | | | | | | | | | |
| LADE | 64.8% | 2.4% | 19.2% | 10.8% | 5.6% | 4.4% | 1.6× | 2.1× | 3.2× | 5.4× |
| REST | 100% | 2% | 95.2% | 76.8% | 52% | 34.4% | 2× | 3× | 4.8× | 8.7× |
| BiLD | 75.2% | 2% | 37.2% | 21.6% | 8.4% | 5.6% | 1.7× | 2.4× | 3.7× | 6.5× |
| **Experiment 3 - Approx Knowledge** | | | | | | | | | | |
| LADE | 13.6% | 4% | 6.8% | 6% | 3.6% | 3.2% | 1.5× | 2× | 3× | 5× |
| REST | 40% | 2% | 29.6% | 20.8% | 14.8% | 8.4% | 1.9× | 2.8× | 4.6× | 8.1× |
| BiLD | 20.4% | 2% | 7.6% | 6.8% | 6% | 3.6% | 1.7× | 2.3× | 3.6× | 6.2× |

Table 2: Attack accuracy under constant size padding (payload within packets padded to 1024 bytes) and variable-length padding (random padding) mitigations. Variable-length padding follows a uniform distribution, $\epsilon \sim Unif(0, D)$, with $D$ from 6 to 48. (Attack: 20 traces per prompt; model temperature: 0.8).

## E  IMPACT OF MITIGATION FOR TOKEN-LENGTH SIDE CHANNEL

In this experiment, we want to measure the impact on our attacks of mitigation against prior attacks, such as token-length side channels (Weiss et al., 2024) that leaks the value of a token based on its associated character length. We evaluate the impact of a mitigation, where every token has been padded to the same size to mask its character length. We measure the attack success rate for our attacks with this mitigation. Note that even with this mitigation, there is variation in the packet sizes, due to the number of tokens contained in each packet, which varies due to speculation. Thus, measuring the packet sizes in this scenario is the same as measuring the token counts.

Figure 10 shows the accuracy for our attack despite this mitigation as temperature varies. At smaller temperatures, the accuracy of our attacks is still significant (with temperature 0.3, in both exact knowledge attacks (experiment 1 and 2), for LADE, the accuracy is about 40-50%, and for BiLD, it is about 80%). At high temperatures, our attack accuracy is reduced: with temperature 1.0, in both the exact knowledge scenarios (experiment 1 and 2), for LADE, the accuracy is reduced to around 20%, and for BiLD, it is reduced to 30%.

Overall, this experiment shows that variations in the tokens counts still enable our attacks despite mitigations against token length side-channels. To fully eliminate our attack, one needs to pad the payload in the packets to the maximum number of tokens per iteration × the maximum token size, which can bloat payload sizes by over 200×.

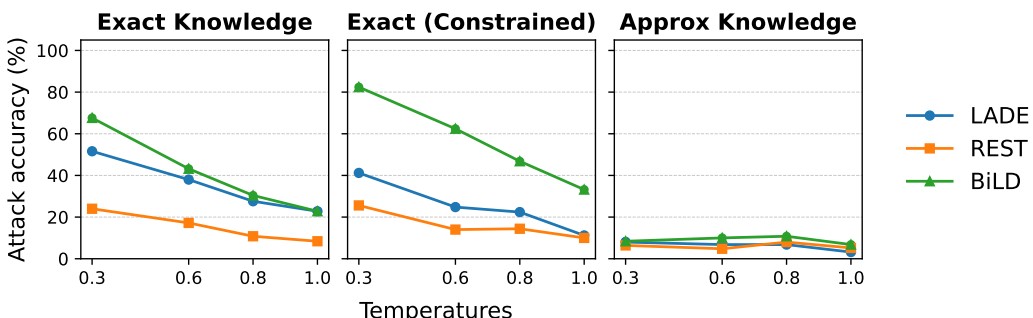

Figure 10: Accuracy of the query fingerprinting attack with each token padded to the maximum size of tokens that can be generated per iteration (1024 bytes)

## F OUT OF DISTRIBUTION TRAINING

### F.1 ATTACK ACCURACY

Table 3 shows the attack accuracy using out of distribution dataset with 50 queries about diseases as training data, and using our original evaluation data set from Experiment 1 as the test set. Our attack achieves an accuracy of 23% to 36% of guessing overlapping symptoms by training wiht the OOD set, achieving significantly higher accuracy than random guessing using OOD set, which achieves only a accuracy of 6%.

| Temperature → | 0.3 | 0.6 | 0.8 |
|---|---|---|---|
| LADE | 25% | 24% | 23% |
| REST | 36% | 35% | 36% |
| BiLD | 25% | 23% | 24% |

Table 3: Accuracy of the Query Fingerprinting Attack with out of distribution training, as the temperature of target model varies from 0.3 to 0.8.

### F.2 DEGREE OF OVERLAP BETWEEN OOD TRAINING SET AND EVALUATION SET

Since we use a 50 class OOD data set (generated using ChatGPT) as training set, and a 50 class data set for the evaluation set (used in Experiment 1), there is no direct 1-1 correspondence between training and evaluation sets. However, artificially, if the symptoms of each disease in the OOD training set has a high overlap with *all* the diseases of the test set, then our attack accuracy can be artificially high. To understand whether this is the case, we study the the overlap in symptoms between diseases in the OOD training set and that in the evaluation set (Experiment 1 - Exact Knowledge).

Figure 13 shows a 2D matrix illustrating symptom overlap between diseases in the OOD training set (X-axis) and diseases in the evaluation set from Experiment 1 (Y-axis). Both sets have 50 diseases. A cell is colored white if the corresponding pair of diseases shares overlapping symptoms. To determine overlap, we query an LLM (e.g., ChatGPT) with the two diseases and check whether they have common symptoms.

In the OOD training set, a small number of diseases (3-4) share symptoms with multiple (5-6) diseases in the evaluation set, and vice versa. However, most training-set diseases overlap with only one or two evaluation-set diseases. Additionally, some evaluation-set diseases have no symptom overlap with any training-set disease; in such cases, an attacker cannot infer the patient's symptoms. This is expected, as the OOD training set is independently constructed from the top 50 diseases most commonly queried on ChatGPT, while the evaluation set is derived from real-world user queries to medical chatbots.

This diversity in overlap patterns reflects a realistic deployment setting. Consequently, the observed attack accuracy of 23%–36% is realistic.

In comparison, a random guessing baseline attack, that assigns a random OOD training-set disease to each evaluation set disease achieves only 6% accuracy, aligning with our observation that most diseases in OOD training set do not overlap in symptoms with many test set diseases. This highlights the potency of our attack using speculative decoding when realistic OOD data is used for training.

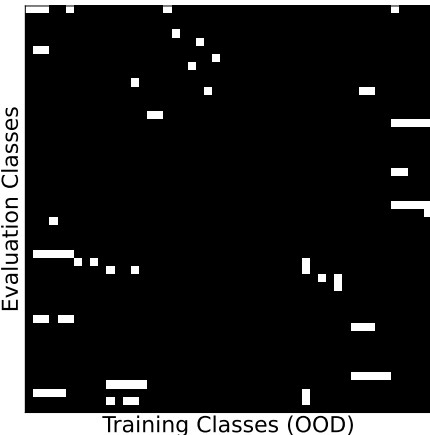

Figure 11: Overlap in symptoms between diseases in the OOD Training set (X-axis) and in the Evaluation set used in Experiment 1 - Exact Knowledge (Y-axis). A white entry indicates that two diseases have similar symptoms (as per GPT-4o), while a black entry indicates no overlap. Overall, there is minimal overlap in symptoms across any random pair of diseases from OOD training dataset and evaluation dataset, representative of realistic OOD datasets.

## G    COMPARING CLASSIFICATION METHODS (RANDOM FOREST, GMM, CNN)

In this section, we evaluate the accuracy of the query fingerprinting attack (Experiment 1 - Exact Knowledge) using different classifiers - Random Forest, GMM and CNN. We study this with for the three speculative decoding techniques LADE, REST and BiLD for a range of temperatures (0.3, 0.6, 0.8, 1), all with traces per query (TPQ) of 30.

For the GMM classifier, we trained the scikit-learn's GMM with 1 component per class, and make predictions via Bayes' Rule. For the CNN classifier, we use three 1D convolution layers with max pooling followed by average pooling and a fully connected layer. For the Random Forest classifier, our setup is similar to Section 4.4.

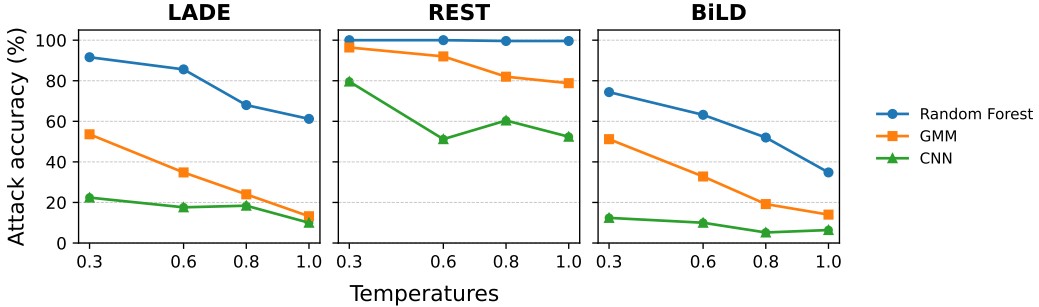

Figure 12: Accuracy of query fingerprinting attack (Experiment 1 - Exact Knowledge) using Random Forest, GMM, and CNN classifiers, for temperatures 0.3, 0.6, 0.8, 1.0 (using 30 traces per query).

As shown in Figure 12, the attack accuracy for the GMM classifier ranges from 13-54% for LADE, 79-96% for REST, and 14-51% for BiLD. However the GMM performs worse than the Random Forest classifier which achieves an accuracy of 61-92% for LADE, 99-100% for REST, and 34-74% for BiLD. While prior work Carlini & Nasr (2024) showed that GMM can achieve query fingerprinting with high accuracy (up to 100%), we observe that their attacks were restricted to 2 classes, contrary to our attacks that classify across 50 classes, in which scenario the GMM achieves lower accuracy and the Random Forest is able to outperform it.

In comparison, the CNN performs worse than both the Random Forest and the GMM, achieving an accuracy of 10-22% for LADE, 52-80% for REST, and 5-12% for BiLD. This is because the training data set is relatively small, with only 250 to 1500 training data points. So we observe that the CNN overfits to a given trace, leading to lower accuracy.

These results lead us to choose Random Forest classifier for all our attacks in this paper, given that it outperforms GMM and CNN.

## H  SCALING BEHAVIOR

In this section, we study how the accuracy of the query fingerprinting attack (with Experiment 1 - Exact Knowledge) with temperature of 0.8 changes as the dataset size (number of diseases) scales from 5 to 50. As shown in Table 4, for BiLD and REST, the accuracy remains relatively stable as the number of diseases increase. The attack accuracy for LADE decreases by about 7 percent when we double the dataset size from 25 diseases to 50. This shows that the attack is relatively robust to scaling of the dataset size.

| | **Number of Diseases in Dataset** | | | | | | | | | |
|---|---|---|---|---|---|---|---|---|---|---|
| | **5** | **10** | **15** | **20** | **25** | **30** | **35** | **40** | **45** | **50** |
| LADE | 92% | 86% | 89.3% | 84% | 76.8% | 74.7% | 74.3% | 73.5% | 74.2% | 68% |
| REST | 100% | 100% | 100% | 100% | 100% | 100% | 100% | 99.5% | 99.6% | 99.6% |
| BiLD | 48% | 40% | 60% | 58% | 45.6% | 48% | 49.1% | 48.5% | 48.8% | 52% |

Table 4: Accuracy of query fingerprinting attack (Experiment 1 - Exact Knowledge) with temperature of 0.8, using 30 traces per query (TPQ) for training as the number of diseases in the dataset scales from 5 to 50.

Next, we study how the training cost scales with the dataset size. Specifically, we measure the cost in terms of the number of traces per query required to reach 90% of the fingerprinting attack (Experiment 1 - Exact Knowledge, with temperature 0.8, and dataset of 50 diseases). We measure this cost as the dataset size scales from 5 to 50 diseases.

As shown in Table 5, we observe that the number of traces (cost) remains relatively stable (for REST) or increases moderately (LADE and BiLD). For REST, this is because the retrieval based speculation provides an extremely stable signal, which allows for high accuracy in the attack with a limited number of traces. In comparison, both LADE and BiLD have more variations, requiring more traces per query as the number of diseases increase.

| | **Number of Diseases in Dataset** | | | | | | | | | |
|---|---|---|---|---|---|---|---|---|---|---|
| | **5** | **10** | **15** | **20** | **25** | **30** | **35** | **40** | **45** | **50** |
| LADE(61.2%) | 3 | 3 | 3 | 4 | 5 | 7 | 7 | 10 | 9 | 9 |
| REST(89.6%) | 4 | 2 | 3 | 4 | 4 | 5 | 5 | 3 | 4 | 4 |
| BiLD(46.8%) | 22 | 30 | 15 | 16 | 28 | 29 | 28 | 27 | 19 | 19 |

Table 5: Number of queries needed to achieve 90% of the fingerprinting attack accuracy (exact knowledge - experiment 1) with temperature of 0.8, as the number of diseases in the dataset scales from 5 to 50.

# I  ABLATION: QUERY FINGERPRINTING ATTACK ON INSURANCE CHATBOT

To show that our attack is not restricted to the medical domain, and also works on other domains, we demonstrate our attack on an insurance chatbot.

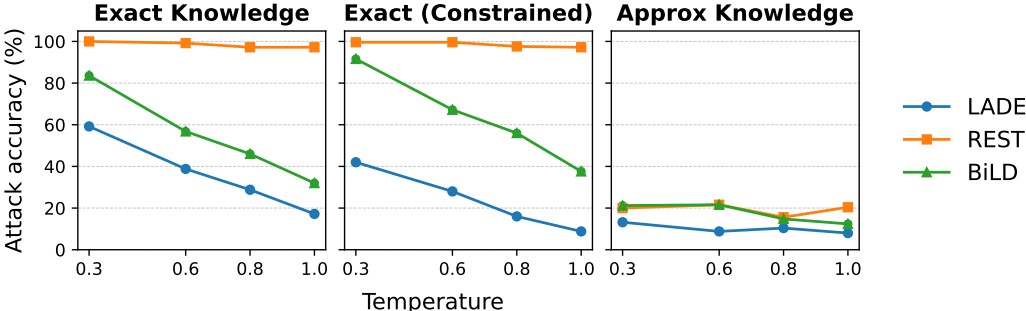

Figure 13: Attack accuracy of query fingerprinting with 50 insurance-related queries as temperatures vary (0.3, 0.6, 0.8, 1.0), using 30 Traces Per Query (TPQ) for training.

We perform the query fingerprinting attack over a set of 50 queries consisting of insurance-related questions, generated by asking ChatGPT for commonly asked insurance-related questions by users. These include questions on health, car, and home insurance. Learning whether a user is asking such questions can allow an attacker to know private information such as whether a user owns a car or a home or has health concerns. We perform the Exact Knowledge, Exact (Constrained) and Approx Knowledge experiments , as designed for the medical chatbot in 4.3. Our REST and BiLD, our attack on an insurance chatbot achieves 97.2–100% and 37.6–91.6% query-fingerprinting accuracy for temperatures 0.3–1.0, slightly outperforming the results we observed on the medical chatbot. LADE with 17.2–59.2% attack accuracy performs worse than in the medical case, however it still outperforms random chance (2%). This shows that the attack does not rely on any medical specific terminologies and can generalize to more diverse domains.

