# OpenReview forum: "When Speculation Spills Secrets: Side Channels via Speculative Decoding in LLMs"
_ICLR.cc/2026/Conference — Submitted to ICLR 2026_

### Official Review · Reviewer_LAHj · 2025-10-26

**Soundness:** 2
**Presentation:** 2
**Contribution:** 3
**Rating:** 4
**Confidence:** 3

**Summary:**

The paper proposes a side-channel attack on speculative decoding, where the adversary can infer the user prompt by observing the packet size sent from a remotely hosted LLM (proxy metric for a per-iteration token counts, i.e., speculation accuracy). For some speculative decoding schemes (e.g., REST), the attack can fingerprint user queries (i.e., can classify the query content out of 50 predefined classes) with near-perfect accuracy. Their method is effective under a real-world deployment scenario using vLLM (Section 4.7). They further discuss that this side channel can be exploited to allow a data extraction attack on an algorithm like REST that relies on a datasource for speculation (Section 5).

**Strengths:**

1. The paper reveals the exploitability of speculation accuracy, which can be estimated in practice based on observing the packet size. It is an interesting and surprising observation that there exists a correlation between speculation accuracy and the input prompt.
2. Their attack essentially works better when the speculation algorithm is more stable (lines 314-316), implying a security-utility tradeoff, and a growing concern as the speculation accuracy improves.
3. They have multiple settings for many experiments (e.g., as described in Section 4.3, lines 240-260), ranging from the easiest setting that gives the upper bound of the attack success, to a more practical setting.

**Weaknesses:**

1. **High-level concern about the contribution**: The method essentially reduces to a random‑forest–based 50‑class classifier (Section 4.4, lines 269–292) that uses “tokens per iteration” as the primary input feature. While reasonable, I believe either of the following must be satisfied/clarified for an acceptable contribution level: (i) technical novelty: include a component specific to the speculative‑decoding setting that materially improves attack success; or (ii) strong practicality: since the current setup assumes a predefined set of 50 classes, I remain unconvinced about the effectiveness in realistic settings where users ask arbitrary topics in arbitrary phrasings.

2. **Detailed analysis of why the attack succeeds**: It is unclear “why” the attack works, i.e., what parts of the trace the classifier relies on when making predictions. The draft presents the method as generic and versatile, yet the experiments are limited to a medical dataset extracted from Han et al. (line 243), so it may be domain‑specific. For example, I hypothesize the correlation between speculation accuracy and user prompt could stem from something specific to disease names, and the method may degrade in other domains e.g., where no technical terms exist. Overall, I would like the authors to go one step deeper so they can claim that the method works for some principled reasons, and to discuss more on when / why it works.

3. **Presentation (minor)**: The typo in the abstract (line 14, transposed “?”) is careless for a conference submission. Also, pasting dozens of raw prompts across 5+ pages (pages 12–18) without formatting/description (e.g., grouping or a table/figure) is unusual and makes the information hard to grasp.

**Questions:**

1. **Concrete setup for OOD ablation** (Section 4.8): The authors conduct a training by using 50 diseases that are generated by GPT-4o as common topics users typically ask about, and then evaluate the prediction accuracy using the 50 predefined classes. Do the 50 classes for training and evaluation have one-to-one correspondence? If yes, how is it different from Experiment 3 (lines 250-260)? If not, how is the evaluation designed?

---

> ### Author Response · Authors · 2025-11-21
> **Responses to Reviewer 4**
>
> **Comment 1: High-level concern about the contribution: The method essentially reduces to a random‑forest–based 50‑class classifier (Section 4.4, lines 269–292) that uses “tokens per iteration” as the primary input feature. While reasonable, I believe either of the following must be satisfied/clarified for an acceptable contribution level: (i) technical novelty: include a component specific to the speculative‑decoding setting that materially improves attack success; or (ii) strong practicality: since the current setup assumes a predefined set of 50 classes, I remain unconvinced about the effectiveness in realistic settings where users ask arbitrary topics in arbitrary phrasings.**
>
> We wish to clarify that our attack is specific to speculative-decoding: the variation in tokens per iteration we use to leak information is introduced specifically by speculative-decoding.
>
> Also, our attack is quite practical when executed on a domain-specific medical chatbot. For such settings, we evaluate a limited set of “50-prompts”, since in many real deployments like healthcare, a constrained prompt space is exposed, such as templated intake forms. An attacker able to enumerate or approximate that space can realistically learn the set of prompts.
> While users can indeed ask arbitrary phrasings, queries are often about diseases or symptoms.
>
> Moreover, in Section 4.8, we show that our attack is still successful without knowledge of the finite set of patient queries. It is possible to learn the symptoms in user’s queries by training the classifier with only out of distribution data (knowledge of prevalent diseases). With this,  we are able to guess symptoms with much higher accuracy (23-35%) than random guessing (2-6%). This can be a serious privacy violation as per the US laws like, HIPAA, which states a health privacy breach occurs whenever "protected health information" (including symptoms), is leaked along with “individually identifiable information” (e.g., IP address of the patient).
>
> **Comment 2: Detailed analysis of why the attack succeeds: It is unclear “why” the attack works, i.e., what parts of the trace the classifier relies on when making predictions. The draft presents the method as generic and versatile, yet the experiments are limited to a medical dataset extracted from Han et al. (line 243), so it may be domain‑specific. For example, I hypothesize the correlation between speculation accuracy and user prompt could stem from something specific to disease names, and the method may degrade in other domains e.g., where no technical terms exist. Overall, I would like the authors to go one step deeper so they can claim that the method works for some principled reasons, and to discuss more on when / why it works.**
>
> Our analysis shows that we can leak patient disease-related details from speculative decoding patterns in medical chatbots, which is a violation of healthcare privacy laws. This is because speculative decoding patterns are by definition “input”-dependent and depend on semantic meanings in phrases within the response. Our classifiers fingerprint these patterns and the underlying semantic meanings in the responses, to leak information.
>
> **Comment 3: Presentation (minor): The typo in the abstract (line 14, transposed “?”) is careless for a conference submission. Also, pasting dozens of raw prompts across 5+ pages (pages 12–18) without formatting/description (e.g., grouping or a table/figure) is unusual and makes the information hard to grasp.**
>
> We have corrected the typo. We provide the raw prompts in the appendix to help with reproducibility, following best practices.
>
> **Question 1: Concrete setup for OOD ablation (Section 4.8): The authors conduct a training by using 50 diseases that are generated by GPT-4o as common topics users typically ask about, and then evaluate the prediction accuracy using the 50 predefined classes. Do the 50 classes for training and evaluation have one-to-one correspondence? If yes, how is it different from Experiment 3 (lines 250-260)? If not, how is the evaluation designed?**
>
> In Section 4.8, the training and test do not have 1-1 correspondence. There is an overlap of only 4 diseases in the two datasets. So this case is different from experiment 3 where there is a 1-1 correspondence. In Section 4.8, the attacker is unable to know the disease queries that the users make to the specific medical chatbot. The attacker uses a list of prevalent diseases (out of distribution for the user interactions with the medical chatbot), trains its classifier based on this OOD set, and then predicts one of the diseases in this OOD set when the traces from the test-set are observed. The accuracy is calculated based on the overlap of symptoms between the predicted disease and the ground-truth, decided by querying another LLM and verified by manual inspection.

---

> ### Comment · Reviewer_LAHj · 2025-11-27
> **Response to authors**
>
> I thank the reviewer for their response. I think several points are not fully clarified:
>
> C1. Now I agree that a 50-class-based experiment is reasonably meaningful. However, in my view, their Experiments 1 and 2 seem toy settings even for the 50-class setting, and the actual practical relevance hinges on Experiment 3 and OOD evaluation. Therefore, I want to reconsider my assessment of their contribution after the point Q1 is further clarified.
>
> C2. If the authors' focus is on the medical domain in particular, I think their analysis is reasonably deep. However, I still believe it would be nicer if they had, e.g., one more setting as an ablation. I remain unconvinced whether the method works outside of the medical domain.
>
> Q1. Now I understand there is no 1-1 correspondence, but I am still struggling to fully understand its design.
> I might be missing something, but based on the current explanation, I think it remains possible in an extreme case that all 50 classes (defined for training) could be considered as correct by any of the 50 classes in the evaluation set (by saying that they have the same symptom). Therefore, the numbers currently being reported in Table 3 seem to have little meaning. Can the authors clarify this correspondence, e.g., by showing the statistics on how many evaluation classes correspond to one training class (or showing the exact correspondence list/graph)?
>
>
> Independently, the authors seem to have clarified in their PDF that their experiments are based on a set of 50 queries. This mitigates the discrepancy between the tone of the paper and its actual experiments. This might have triggered my feeling about C1, so I think this edit was appropriate.

---

> ### Author Response · Authors · 2025-11-30
> **Response to Reviewer LAHj**
>
> Thank you for reviewing our rebuttal and providing a response! Your suggestions are very helpful. We have addressed both your remaining concerns & questions below.
>
> **C1/Q1: Can the authors clarify the correspondence between OOD training and evaluation datasets? e.g.,  show the statistics on how many evaluation classes correspond to one training class?**
>
> We analyzed the overlap in symptoms between the 50 diseases in the OOD Training set and the 50 diseases in the  evaluation set. We have added Figure-11 in Appendix F.2 to provide a 2D matrix plotting this overlap, where a white entry indicates a pair of diseases between Training and Evaluation have similar symptoms. Overall, there is minimal overlap in symptoms across any random pair of diseases from the OOD training dataset and evaluation dataset, with each evaluation disease overlapping with 1-2 training diseases. Some diseases do not have overlap, while 3-5 diseases have overlap with 4-5 others. Thus, our training set is quite representative of a realistic OOD dataset. When we attempt a random guessing attack, using a random OOD training set disease as a guess for each diseases in the evaluation set, the accuracy is only 6%, whereas our attack using OOD data and speculative decoding based traces has 23% to 36% accuracy, demonstrating the capability of our attack.
>
> Please see our newly added Appendix F.2 and Figure 11 for more details.
>
>
> **C2. Can you show one more setting as an ablation? Does the method work outside of the medical domain?**
>
> To show that our attack indeed works outside the medical domain, we demonstrate our attack on a chatbot in the insurance domain.
>
> We perform our query fingerprinting attack with 50 queries regarding insurance questions, generated by asking ChatGPT for commonly asked insurance questions. These include questions on health, car, and home insurance. For REST, BiLD and LADE our attack on an insurance chatbot achieves 97–100%,  38–92% and 17-59%  attack accuracy for temperatures 0.3–1.0, in a similar range or slightly outperforming our results on a medical chatbot. This shows our attack does not rely on any medical-domain specific information, and works on other domains.
>
> We have added these results in Appendix I and Figure 13.

---

### Official Review · Reviewer_KSN3 · 2025-10-28

**Soundness:** 2
**Presentation:** 3
**Contribution:** 2
**Rating:** 4
**Confidence:** 3

**Summary:**

This paper considers information leakage in speculative decoding for LLMs. It proposes an attack based on packet sizes and demonstrates that it can identify user queries with high success rates across four speculative decoding schemes in specific medical query scenarios. It proposes several defense mechanisms, which can effectively reduces the attack success rates at high costs.

**Strengths:**

1. The paper is mostly well-written and easy to follow.
2. It is an interesting observation that per-iteration token count or packet size can leak private information.
3. The attack works across different speculative decoding schemes.
4. The defense mechanisms can effective reduce the risk of information leakage.

**Weaknesses:**

1. The experiments is limited to a very special medical chatbot scenario where there are only a small number of diseases. There is no experiment about scaling up the number of possible labels, or diseases in this case.
2. The proposed defense mechanisms are all very costly and not very practical.

**Questions:**

1. How will the attack success rate scale with the number of diseases?
2. How will the attack cost, e.g. in terms of number of profiling examples and training cost,  scale with the number of diseases?
3. In the out-of-distribution experiment, how different are the two distributions? In particular, how many of the "50 common diseases" overlap with the training examples? Since the attack success rate already drops significantly in the current experiment, should one expect the risk to be of little practical importance when the number of diseases is large?

---

> ### Author Response · Authors · 2025-11-21
> **Responses to Reviewer 3**
>
> **Comment 1: The experiment is limited to a very special medical chatbot scenario where there are only a small number of diseases. There is no experiment about scaling up the number of possible labels, or diseases in this case.**
>
> We evaluate possible attacks on domain-specific chatbots like medical chatbots.  In these cases, the set of queries that can be answered are by definition constrained, leading us to develop experiments with a closed world setting by default (Experiments 1 and 2). The goal of Experiment 3 (semantically similar but non-identical) is to relax this assumption and  evaluate training sets where the attacker knows the diseases in the test set, but not the exact user queries. Finally Section 4.8 approximates a totally out of distribution data set for training, extracted based on prevalence information of diseases which is often public.
>
> Since curating real-world datasets with more diseases is beyond the scope of this work, we show scalability of the attack success and attack costs, while using only a subset of the diseases to understand the trends.
>
> **Comment 2: The proposed defense mechanisms are all very costly and not very practical.**
>
> Indeed, our attack is challenging to mitigate in practice: both packet padding and aggregation introduce non-trivial overhead or latency. This underscores the significance of our findings regarding the attack! The goal of this paper is to highlight the privacy risks introduced by attacks exploiting speculative decoding and to analyze the trade-offs of existing mitigations. If published, future research can then focus on more efficient and practical defenses.
>
>
> **Question 1: How will the attack success rate scale with the number of diseases?**
>
> We evaluated the accuracy of the query fingerprinting attack (with Exact Knowledge) with T=0.8. changes as the dataset size (number of diseases) scales from 5 to 50. The results are provided  in Appendix H (Table-4).  For BiLD and REST, the accuracy remains relatively stable as the number of diseases increase. The attack accuracy for LADE decreases by about 7 percent when we double the dataset size from 25 diseases to 50. This shows that the attack is relatively robust to scaling of the dataset size.
>
> **Question 2: How will the attack cost, e.g. number of profiling examples and training cost, scale with the number of diseases?**
>
> We evaluate the training cost as the dataset size in Appendix H (Table-5). Specifically, we measure the cost as the number of traces per query required to reach 90% of the fingerprinting attack (Exact Knowledge, with T=0.8, and a dataset of 50 diseases). We measure this as the dataset scales from 5 to 50 diseases and provide the results in Table 5.
>
> We observe that the number of traces (cost) remains relatively stable (for REST) or increases moderately  (LADE and BiLD). For REST, this is because the retrieval based speculation provides an extremely stable signal, which allows for high accuracy in the attack with a limited number of traces. In comparison, both LADE and BiLD have more variations, requiring slightly more traces per query as the number of diseases increase.
>
>
> **Question 3: In the out-of-distribution experiment, how different are the two distributions? In particular, how many of the "50 common diseases" overlap with the training examples? Since the attack success rate already drops significantly in the current experiment, should one expect the risk to be of little practical importance when the number of diseases is large?**
>
> There are only 3 or 4 diseases that appear in both datasets. To quantitatively demonstrate that this overlap is insignificant, we use this OOD dataset to perform random guesses for our attack on our test-set prompts. The attack accuracy with such a random guess is only 6%, much lower than our attack rate of 23%-36% by leveraging this OOD data for creating speculation traces for training.
>
> Practical importance. In domains such as medical systems, even a single instance of unauthorized disclosure of Protected Health Information (e.g., disease suffered by a patient)  is a violation of healthcare privacy laws (e.g., HIPAA in the USA). We clearly demonstrate our attack accuracy is much higher than random guessing, highlighting the practical risk of medical chatbots using widely adopted LLM optimizations like speculative decoding.

---

### Official Review · Reviewer_QPdf · 2025-10-31

**Soundness:** 3
**Presentation:** 2
**Contribution:** 3
**Rating:** 6
**Confidence:** 4

**Summary:**

This paper reveals that speculative decoding techniques used to accelerate inference in large language models pose severe privacy risks. By analyzing packet-size patterns in encrypted network traffic, attackers can infer whether internal speculations succeed or fail, enabling them to identify users’ sensitive queries or to extract the confidential parameters that drive speculation.

Experiments across multiple speculative-decoding schemes and real-world deployment settings confirm the effectiveness of this side-channel attack. Although defenses such as packet padding or token aggregation exist, they typically force a trade-off between performance and privacy.

**Strengths:**

1. This paper is the first to reveal a packet-size-based side-channel attack introduced by speculative decoding techniques in LLMs.   It explicitly differentiates this work from prior LLM side-channel attacks, such as token-length leakage  and timing attacks by focusing on input-dependent speculation patterns.

2. The attack is validated across four speculative decoding schemes (REST, LADE, BiLD, EAGLE) and tested in both academic prototypes and the production-grade vLLM serving framework, confirming its novelty as the first systematic exploration of this specific side channel .

3. The paper’s experiments are comprehensive, as evidenced by the design of its fingerprinting attack and multi-dimensional evaluation.

**Weaknesses:**

1. There are still issues with writing and typesetting. For example, the caption of Figure 1; the font size in Figure 5 is excessively small; and in tables (e.g., Table 1, Table 4), the layout is overly compact.

2. Although Experiment 3 (semantically similar but non-identical queries) and Section 4.8 (out-of-distribution training) evaluate the fingerprinting attack under approximate or out-of-distribution dataset setups, both configurations remain somewhat idealized.

3. The paper provides no justification for choosing random forest over other advanced methods and does not explore whether using more sophisticated algorithms could reveal higher attack potential or more robust speculation patterns.

4. While token aggregation is shown to reduce attack accuracy, the paper does not measure its impact on end-user perceived latency

5. Common LLM optimizations like paged attention split the KV cache across multiple GPUs, which may interact with token aggregation to alter packet generation logic .     The paper does not test this interaction, so it remains unknown if token aggregation is broken or weakened by paged attention.

**Questions:**

1. Intuitively, the traffic pattern of each prompt–response pair seems likely to be unique. Would it be possible to introduce additional metrics and perhaps specific thresholds to make the "same/different" distinction more explicit?
2. Could concurrent traffic affect the attack accuracy? Have any experiments measured the fingerprinting attack’s performance under real Internet conditions?
3. Could employing other classifier algorithms improve the overall effectiveness to some extent?
4. The current dataset appears modest in scale, and the prompts are short and straightforward. While this nicely demonstrates the attack under ideal conditions, I wonder how it would behave when the scenarios grow more complex.
5. The attack’s success rate varies across decoding strategies. Is there any theoretical insight that helps explain why these differences emerge?

---

> ### Author Response · Authors · 2025-11-21
> **Responses to Reviewer 2**
>
> **Comment 1: Writing and Typesetting Issues.**
> Thanks. We have fixed these in the revision.
>
> **Comment 2 / Q4: Experiments 3 and Section 4.8 use OOD or approximate setups, but are still idealized. How would the attack perform in more complex scenarios?**
>
> We evaluate attacks on domain-specific chatbots like medical chatbots, where queries are naturally constrained, motivating closed-world Experiments 1–2. The goal of Experiment 3 (semantically similar but non-identical) is to relax this assumption and evaluate training sets where the attacker knows the diseases in the test set, but not the exact user queries. Finally Section 4.8 approximates a totally out of distribution data set for training, extracted based on prevalence information of diseases which is often public.  Indeed the next step is to explore open-world data leakage through speculation side channels, which we leave for future work.
>
> **Comment 3 / Q3: Justification for random forest over other methods?**
>
> We tried both GMM and CNN; both perform worse compared to Random Forest classifier (the default we used).  In Appendix G, we add the accuracies with GMM and CNN.
>
> GMM’s attack accuracy reaches a max of 54% for LADE, 96% for REST, and 51% for BiLD. CNN performs worse than GMM, with a max accuracy of 22% for LADE, 80% for REST, and 12% for BiLD. Random Forest outperforms both GMM and CNN, with a maximum accuracy of 92% for LADE, 100% for REST, and 74% for BiLD.
>
> In Section 4.7, we added evaluations of inter-arrival time channel (Carlini and Nasr, 2024) using GMM (like in prior work). Even with GMM, our packet size channel outperforms prior work. Note that we use 50 classes and prior work used only 2 classes - hence accuracies we report for prior work are much less than 100% as they reported in their paper.
>
> **Comment 4: While token aggregation is shown to reduce attack accuracy, the paper does not measure its impact on end-user perceived latency.**
>
> Token aggregation linearly increases the inter-arrival times for packets, linearly increasing the latency between text renderings for a user. We have clarified this in Section 6.1. Measuring the impact of this on user perception is hard without user studies, which is beyond the scope of this paper.
>
> **Comment 5: Is token aggregation broken by paged attention (splitting KV Cache)? **
>
> Token aggregation we propose is a network-layer or application-layer policy that can buffer and coalesce tokens in packets before transmission. This is orthogonal to GPU kernel-level optimizations like paged-attention or KV cache splitting, that are implemented at lower levels of the software stack.
>
> **Q1: The traffic pattern of each prompt–response pair seems to be unique. Additional metrics or specific thresholds to make the "same/different" distinction explicit?**
>
> In Section-4.2 (Lines 220-228) we already suggest how to quantitatively identify unique prompt/response pairs, and make the “same/different” distinction explicit.
>
> The token-per-iteration patterns across multiple repetitions are largely unique and reproducible for each prompt-response pair (Figure 4, Appendix B). To quantify this, we generated two traces per prompt for 50 prompts from the MedAlpaca dataset (Han et al., 2023) at temperature 0.3 using LADE and computed pairwise cosine similarity. Traces for the same prompt/disease scored 0.9–1, while different prompts scored 0.4–0.8, showing our traces are unique to fingerprint prompt-response pairs. Based on these quantitative measurements, we use a threshold of 0.85: any two traces with cosine similarity score higher than 0.85 are classified as the same prompt-response pair.
>
> **Q2: Does concurrent traffic affect attack accuracy? Any experiments measured over real Internet conditions?**
>
> Section 4.7 already shows the experiment with vLLM + EAGLE running on a remote server thousands of miles from the client, communicating over the internet. We achieve accuracies of 24-78% across temperatures of 1 to 0.3.
>
> As shown in Section 4.7, our attack is not affected much by concurrent traffic, as we measure the tokens per iteration based on sizes of packets originating from the LLM server, unlike prior work (Carlini and Nasr, 2024) which measures the inter-arrival times between LLM server packets, which are considerably impacted by concurrent traffic (accuracies drop to almost random guess 2% for prior work with high server or network load).
>
> **Q5: Theoretical insight why the attack’s success varies across strategies?**
>
> We describe the insight in lines 315 to 317.
>
> REST has higher attack accuracy than BiLD and LADE, as REST's retrieval-based speculation provides deterministic speculation patterns that are highly correlated with phrases in the response holding semantic information (even at higher temperatures). BILD and LADE are both more probabilistic, resulting in lower accuracies especially at higher temperatures.

---

### Official Review · Reviewer_mQbv · 2025-10-31

**Soundness:** 3
**Presentation:** 2
**Contribution:** 3
**Rating:** 6
**Confidence:** 4

**Summary:**

This paper studies side-channel attacks on speculative decoding in LLMs that can leak information about a user’s prompt. Different prompts/responses can lead to unique patterns of accepted and rejected draft tokens, which can be manifested as differences in packet sizes (if the response is streamed). Therefore, a network attacker can observe packet sizes to predict information about a user’s prompt.

The paper runs experiments in controlled, simulated settings, finding that the attack can accurately predict at low temperatures when the exact set of test prompts is known beforehand. At higher temperatures, or when the exact test prompts are not known beforehand, accuracy significantly decreases, but is still better than random guessing. The paper also proposes and evaluates defenses to mitigate the attack.

**Strengths:**

It is important to draw attention to the potential privacy risks of LLM inference techniques, given how widely LLMs and inference optimizations are used today. The paper runs simple, proof-of-concept experiments in controlled, simulated settings that demonstrate that speculative decoding can leak information about user prompts via packet sizes. The paper also proposes and evaluates defenses against the attack, giving concrete mitigations that LLM providers can implement.

Code and documentation are uploaded in the supplementary material, which enhances the reproducibility of the paper.

**Weaknesses:**

* The high accuracies reported in the abstract are only achieved in limited settings: low temperature (0.3) and the exact set of 50 test prompts are known and used at training time. When the temperature increases, or when the exact test prompts are not trained on, the accuracies decrease significantly, although still above random guessing. It seems like much of what the attack is doing is memorizing the fingerprint for specific responses, as indicated by the brittleness to increasing temperature.

  To avoid being misleading, the accuracies in the abstract should be updated or omitted, or the caveat of the limited setting should be explicitly stated.
* No experiments are run on real-world production systems like ChatGPT or Claude, which are done by the related works [Weiss et al. (2024)](https://arxiv.org/abs/2403.09751) and [Carlini et al. (2024)](https://arxiv.org/abs/2410.17175). This would provide more evidence of the efficacy of the attack in more realistic settings, where more factors are unknown (speculative decoding implementation, streaming logic, etc.)
* I think that the tables would be more naturally presented as graphs. The tables generally show how the accuracy changes as some parameter changes (temperature, traces per query, etc.). For example, Table 2 could plot accuracy against temperature, with one line representing each speculative decoding method. The tables have many numbers, making them hard to read and see the overall trends.
* The formatting of the paper could use some polishing. For example, parenthetical citations are not correctly formatted throughout the paper. The parentheses are missing, so they interrupt the sentence and make them harder to read.

**Questions:**

### Questions

1. For the locally run experiments, how are the packet sizes determined? How is it determined when to send each packet, and how many tokens are sent in each packet?
2. When the attack accuracy is high, how similar are the responses/traces at test time compared to the ones from training time? Is the model giving the same response, or is there more diversity?
3. For REST, how similar are responses to each other as the temperature increases? Since REST is retrieval-based, I am wondering if increasing temperature does not increase response diversity as much as in the other speculative decoding methods, leading to the high accuracy for REST.
4. Do you have a reference or explanation for modeling times in high server load with a log-normal distribution?
5. In the out-of-distribution experiment (Section 4.8), for each ground truth prompt, there may be several diseases that have similar symptoms. So, random guessing would have a higher performance than normal. What is the accuracy achieved by random guessing under this evaluation?
6. In the datastore leakage attack (Section 5), what is the false positive rate, i.e., classifying a sequence as in the datastore when it does not actually appear?
7. How does the packet size increase 230x when padding to 1024 bytes? This means that the original packet size is just 4 bytes, which is enough room for at most 4 characters (not even counting metadata).
   1. Also, the packet size can be made smaller, and if there are too many tokens generated in one iteration, it can just be split into several packets. There can be some delay between these packets so the attacker cannot tell that it was one iteration.
8. From the code in the supplementary material, it looks like a random forest is also used when evaluating the inter-arrival time side-channel attack (Carlini & Nasr, 2024). However, that paper uses Gaussian Mixture Models and convolutional neural networks for predicting prompts/topics from the timing data. For a fair comparison, GMMs should be used for the time side-channel, as it may achieve better performance.
9. More generally, did you try any methods other than random forests? Other methods may perform better on both the packet size and inter-arrival time data.
10. Do real-world production systems such as ChatGPT, Claude, etc. actually send multiple tokens per packet? Are there systems that always send exactly one token per packet?

### Notes

1. \> appears as ¿ in the abstract.
2. Parenthetical citations are missing the parentheses throughout the paper. `\citep` should be used.
3. There is an extra indentation at the start of lines 142 and 154\.
4. Line 157 typo: “wand” should be “and”
5. It would be good to have more descriptive titles for each experiment in addition to just “Experiment 1”.

---

> ### Author Response · Authors · 2025-11-21
> **Responses to Reviewer 1**
>
> **Comment-1: Accuracies in abstract only hold for low temperature (0.3) and known set of 50 prompts; seems like memorizing fingerprints. Should update or note limitations.**
>
> Even at higher temps, accuracy much higher than random (70% at T=0.3, 40% at T=0.6, 20% at T=1.0 vs 2% for random), indicating leakage of semantic information rather than memorized responses. Thanks for the suggestion. We have updated the abstract to state temperature and the exact setting under which the accuracies are achieved.
>
> **Comment-2/Q10: No tests on real systems like ChatGPT/Claude. Do they send multiple tokens per packet?**
>
> Prior work shows attacks on ChatGPT/Claude but given their black box nature, it is hard to reason why the attack works. We focus primarily on academic speculative decoding techniques to confirm the root-cause of the issue: implying that any service using speculation is likely vulnerable. Our deployment of vLLM on a remote server indeed mimics a real-world deployment, given that it is commonly used for open-source LLM deployments. Using WireShark, we confirmed that ChatGPT in fact sends one token per packet. To model this behavior, in our vLLM + Eagle implementation in Section 4.7 , we model multiple tokens generated per iteration in separate packets, sent immediately one after the other.
>
> **Comment-3/4: Add Graphs. Parenthetical Citations.**
>
> Thanks! We have added graphs (Figures 3, 4 ,6, 7, 10). And updated parentheses for citations.
>
> **Q1: For local experiments, how are packet sizes set and when are tokens sent?**
>
> Each iteration sends a packet per iteration with all tokens generated that round (1 to N). Attacker infers # of tokens via packet size (tokens are fixed size).
>
> **Q2: When high accuracy, is there diversity in model responses?**
>
> Yes. Even at T=0.3, the responses differ from run to run. The traces also differ but maintain identifiable shifted patterns (likely tied to semantic patterns) causing leakage . Only at T<0.1 do responses converge.
>
> **Q3: Are responses diverse at high temperatures for REST?**
>
> REST is lossless and maintains diversity in response like baseline LLM at high temperature. Its retrieval-based speculation yields a stable signal even when responses significantly vary, which is the root cause of the high accuracy.
>
> **Q4: Reference for log-normal inter-arrival times?**
>
> Benson et al. (SIGCOMM’10) shows datacenter packet inter-arrivals follow log-normal under high load. We have added this in the revision.
> Benson T, Anand A, Akella A, Zhang M. Understanding data center traffic characteristics. ACM SIGCOMM Computer Communication Review. 2010 https://dl.acm.org/doi/10.1145/1672308.1672325
>
> **Q5: Random guessing accuracy in OOD setting?**
>
> We modeled a "Sophisticated” random guessing using OOD diseases as suggested. For each test set disease, we guess a random disease from the OOD set. This gives us an accuracy of 6% (compared to previous random guess accuracy of 2%). This is still less than 23-35% accuracy with our classifier trained using OOD. We have added this in Section 4.8.
>
> **Q6: False positives in datastore leakage?**
>
> We do not have false positives for datastore leakage. As a locally hosted model, we reveal exact token counts from REST. And REST only provides extra tokens per iteration, if these tokens deterministically exist in its datastore. Hence there are no false positives in the datastore leakage.
>
> **Q7: Why 230× overhead with 1024-byte padding? Packet Splitting mitigation?**
>
> This ratio only counts the payload (metadata excluded), since the metadata bytes can vary based on protocol/inference framework. This provides a worst-case estimation of the payload size increase. We added this clarification in Section 6.1
> Splitting packets can help as a mitigation, but attackers can also infer the token count by tracking packet count per iteration. A robust defense also needs to limit packets per iteration, thus adding latency. We have added a discussion of this in Section 6.1.3.
>
> **Q8–9: Why Random Forest, not GMM/CNN?**
>
> We tried both GMM and CNN; both perform worse compared to Random Forest classifier (the default we used).  In Appendix G, we add the accuracies with GMM and CNN.
>
> GMM’s attack accuracy reaches a maximum of 54% for LADE, 96% for REST, and 51% for BiLD. CNN performs worse than GMM, with a max accuracy of 22% for LADE, 80% for REST, and 12% for BiLD. Random Forest outperforms both GMM and CNN, with a maximum accuracy of 92% for LADE, 100% for REST, and 74% for BiLD.
>
> In Section 4.7, we added the evaluation of inter-arrival time channel (Carlini and Nasr, 2024) using GMM (like in prior work). Even with GMM, our packet size channel outperforms prior work. Note that we use 50 classes and prior work used only 2 classes - hence the accuracies we report for prior work are much less than 100% as they reported in their paper.

---

### Author Response · Authors · 2025-11-27
**Response to Rebuttals**

Hello,
Since we haven't heard back a response to our rebuttal, we just wanted to check in and see if the reviewers had a chance to read through our rebuttals, and whether they had any questions. We will be happy to answer them!

We have also uploaded a revised paper incorporating all the changes requested by the reviewers, with changes highlighted in blue.

~ Authors

---

### Author Response · Authors · 2025-11-30
**Final Summary of Rebuttal and Revision**

We thank the reviewers and the AC for their time and constructive feedback. Below, we briefly summarize the paper’s core contributions and the changes made in the paper to address reviewer concerns (changes highlighted in blue in the revision).


## Main Contributions

**Novelty.** This work is the first to identify speculative decoding (SD) as a privacy side channel in large language models. We show that SD introduces input-dependent variance in accepted token counts, which translates into variable packet sizes observable to a network adversary. This leaks semantic information about user queries, opening a new attack surface at the intersection of LLM systems and security.

**Significance.**  In a medical chatbot, where users ask queries about symptoms/diseases, our attacks leak user prompts (diseases) in closed-world settings (with 50 queries) with 78–100% accuracy,  and 23-36% accuracy when true queries are unknown but approximated using out-of-distribution (OOD) data, far exceeding random guessing (2-6% accuracy). We demonstrate this across four SD methods: REST, LADE, BiLD, and EAGLE.

**Practicality and Real-World Relevance** As in prior work (Weiss et al.), we show that attackers controlling routers or malicious ISPs can capture encrypted packet sizes via Wireshark, enabling our attack. Under HIPAA, leakage of user-linked symptom information constitutes a privacy violation; our query fingerprinting attack leaks such user symptom  information with far higher accuracy than random guessing, making the threat practical and regulatorily significant.

**Robustness & Comparison to Prior Work.** We validate the attack in a real-world deployment with a vLLM server running SD technique EAGLE, located over 1,000 miles from the client. Over the internet, our packet-size–based attack achieves 24–78% accuracy, significantly outperforming prior attacks (Carlini & Nasr, 2024) based on inter-arrival-time (4–27% accuracy), while also exhibiting greater resilience to network noise and server-load variation compared to prior work.

**Defenses.** We evaluate mitigations such as packet padding and token aggregation, showing that while defenses are possible, they incur non-trivial costs. Highlighting these trade-offs helps guide future research on secure LLM deployment.

## Changes to Address Reviewer Concerns

**Choice of Classifier (Random Forest vs. GMM/CNN).**
* We justify our use of Random Forest by comparing it against GMM and CNN classifiers. Our Random Forest classifier achieves higher maximum accuracy (74-100%) compared to GMM (54-96%) and CNN (12-80%). (**Appendix G**)
* We also compare against prior work (Carlini & Nasr, 2024), using their GMM-based approach, as requested by one reviewer, and show that our attack still outperforms theirs regardless of whether Random Forest or GMM is used. (**Table-1**)

**Clarifications on Out-of-Distribution (OOD) Experiments**
* To evaluate attack feasibility when user queries are unknown, we train on an OOD dataset of 50 common diseases (Appendix A5) and test on a disjoint set of 50 real-world medical queries (Appendix A1). To address concerns about overlap in these two datasets, we include a symptom-overlap matrix showing minimal overlap between training and evaluation classes (most evaluation diseases overlap with only 1–2 training diseases in symptoms, some with none) (**Appendix F.2, Figure 11**)
* We further include a control experiment using a “sophisticated” random-guessing baseline, where a random OOD disease is used to guess an evaluation set disease. This random guess attack achieves only 6% accuracy, compared to 23–36% accuracy using our OOD-trained classifier. (**Section 4.8**)

**Generalizability & Scaling Behavior**
* To show the attack is not medical-domain–specific, we demonstrate a similar query fingerprinting attack on an insurance chatbot (with queries on home, car and health insurance), achieving 17–97% accuracy and leaking personal attributes (e.g., car vs. home ownership)  (**Appendix I, Figure 13**)
* To analyze scalability, we vary the number of classes from 5 to 50 and measure both attack accuracy and profiling cost (traces needed to reach 90% of final accuracy). Accuracy remains stable with at most a 7% drop, while profiling cost increases modestly and sub-linearly  (**Appendix H, Tables 4–5**)

**Improvements to Presentation and Writing**
* We replaced several tables with figures for improved clarity, per reviewer suggestions  (**Figures 3,4,6,7**)
* We clarified the abstract and mitigation (**Section 6.1**) as per reviewer feedback to contextualize results, and corrected typos throughout the paper.

---

### Meta-Review · Area_Chair_WAWk · 2025-12-29

**Summary:**

This work reveals a packet-size-based side-channel attack introduced by speculative decoding (SD) techniques in LLMs. It systematically validates the attack across four SD methods (REST, LADE, BiLD, EAGLE). Notably, it demonstrates high practical relevance in medical chatbots (78–100% closed-world accuracy) and evaluates mitigations, highlighting their performance trade-offs.

After the rebuttal, the common concerns regarding generalizability, methodological justification, experimental clarity, and insufficient real-world relevance partially remain. Besides, there are minor issues like suboptimal presentation (tables, typos).

**Reviewer Concerns:**

The remaining concerns include

- No tests on commercial LLM systems (only open-source vLLM), leaving attack feasibility in black-box production environments uncertain.

- As the authors said, token aggregation linearly increases the inter-arrival times for packets.

**Reviewer Scores:**

Given the above concerns, the reviewers with a score of 4 may not increase the score.

---

### Decision · Program_Chairs · 2026-01-26

Reject